# Reviving Life on the Edge: Joint Score-Based Graph Generation of Rich Edge Attributes

**Nimrod Berman**
*Bosch, Ben-Gurion University*

**Eitan Kosman**
*Bosch*

**Dotan Di Castro**
*Bosch*

**Omri Azencot**
*Ben-Gurion University*

**Reviewed on OpenReview:** *https://openreview.net/forum?id=pxdSm7PW5Q*

## Abstract

Graph generation is integral to various engineering and scientific disciplines. Nevertheless, existing methodologies tend to overlook the generation of edge attributes. However, we identify critical applications where edge attributes are essential, making prior methods potentially unsuitable in such contexts. Moreover, while trivial adaptations are available, empirical investigations reveal their limited efficacy as they do not properly model the interplay among graph components. To address this, we propose a joint score-based model of nodes and edges for graph generation that considers all graph components. Our approach offers three key novelties: **(1)** node and edge attributes are combined in an attention module that generates samples based on the two ingredients, **(2)** node, edge and adjacency information are mutually dependent during the graph diffusion process, and **(3)** the framework enables the generation of graphs with rich attributes along the edges, providing a more expressive formulation for generative tasks than existing works. We evaluate our method on challenging benchmarks involving real-world and synthetic datasets in which edge features are crucial. Additionally, we introduce a new synthetic dataset that incorporates edge values. Furthermore, we propose a novel application that greatly benefits from the method due to its nature: the generation of traffic scenes represented as graphs. Our method outperforms other graph generation methods, demonstrating a significant advantage in edge-related measures.

## 1 Introduction

Generative modeling is a persistent challenge in scientific and engineering fields, with broad practical use cases. The primary goal is to understand a large database's inherent distribution, enabling new samples to be generated. In many use cases, using a graph representation is convenient for describing the samples, such as in molecule and protein design (Du et al., 2022), neural architecture search (Oloulade et al., 2021), program synthesis (Gulwani et al., 2017), and more (Zhu et al., 2022).

The exploration of generative modeling is a longstanding endeavor, marked by the development of various methodologies throughout the years, including variational autoencoders (Kingma & Welling, 2014), adversarial learning (Goodfellow et al., 2014), normalizing flows (Rezende & Mohamed, 2015), and diffusion models (Sohl-Dickstein et al., 2015). These approaches have been used to generate various information types

(Dhariwal & Nichol, 2021; Naiman et al., 2024b;a) and solve many downstream tasks (Ho et al., 2022; Naiman et al., 2023; Berman et al., 2024). Recently, the generation of graph data has gained increased attention (Li et al., 2018; You et al., 2018). In particular, the modeling of graph distributions through score-based approaches (Song & Ermon, 2019) stands out as a promising avenue that necessitates a deeper investigation.

Generally, a graph contains several components that have mutual dependencies. One is the nodes, a set of entities with possibly assigned attributes. The second is the adjacency information that specifies the nodes' connectivity with potentially assigned features. For example, one could use this structure to model a molecule, where nodes represent atoms with atomic numbers as their attributes, and the adjacency information represents the intramolecular bonds and their types. The involvement of several components in the graph whose attributes have a mutual interplay introduces the challenge of modeling the components altogether along with their relations. In order to address this challenge, a discrete diffusion approach for generating categorical node and edge attributes was proposed (Vignac et al., 2023). However, extending it to sampling real-valued attributes remains non-trivial. Yet, while a discrete diffusion process fits well in certain cases, we advocate in this work the consideration of a more general problem with continuous score-based frameworks.

Recent works on score-based models for graph generation have made significant strides but remain limited in scope (Niu et al., 2020; Jo et al., 2022; Fan et al., 2023). While proficient in their designated tasks, these models either completely exclude edge attributes or treat graph components separately, limiting their capacity to capture complex relational structures. We address these limitations with a unified framework that enables the generation of node and edge attributes. An exemplary task with dominant edge attributes is shown in Fig. 1. This task involves translating a driving scene into a graph representation, similar to the approach in VectorNet (Gao et al., 2020). One prominent feature is the existence of edge attributes that introduce relative and interactional information between the road participants. For instance, an edge feature like a "looking at" flag–which indicates if one vehicle is actively observing another–captures situational awareness crucial in dense traffic. This relational cue, such as mutual acknowledgment during lane changes, is necessary for accurately modeling driving behavior but emerges only from the interaction itself, making it impossible to derive from node features alone. This and less expressive versions of such representation are prevalent in motion prediction literature (Huang et al., 2022), with models utilizing it often leading the prediction task leaderboards (Caesar et al., 2020). Consequently, we are interested in generating scenes for this type of representation. An additional natural example originates from Markov Decision Processes (MDPs), where edge features like transition probabilities and rewards capture the dynamics of state transitions. Without edge features, this information would have to be redundantly stored at the node level, increasing complexity and obscuring transition flow. Encoding these properties directly on edges provides a compact, interpretable representation that preserves the MDP's natural structure.

The closest evidence to graph generation with attributes appears in "Attributed Graph Generation" (Pfeiffer III et al., 2014; Lemaire et al., 2024). However, this formulation only associates attributes with the nodes, leaving the edges free from additional attributes besides the adjacency matrix. As we find critical tasks where edge attributes are dominant, we want to promote awareness of this problem to enable the generation of graphs with richer information and more expressiveness than the current formulation of attributed graphs. Consequently, in this study, we leverage the evident insight that edge attributes convey neighborhood information and provide instrumental data absent in the adjacency matrix crucial for generating edge attributes. We propose to encode graph distributions via a joint Stochastic Differential Equation (SDE), describing the evolution of node and edge attributes. Importantly, our technique jointly solves for graph elements. Consequently, it benefits from the synergetic connections between nodes and edges. In comparison, GDSS (Jo et al., 2022) proposed a similar diffusion system for adjacency and nodes that opt for a separated solution that may be sub-optimal in encoding certain graphs. We solve our joint SDE by further strengthening the dependencies between nodes and edges. In practice, this is achieved by combining node, edge, and adjacency information in an attention module, maximizing the mutual interplay of the graph components. Overall, our approach is designed to maximally exploit the information encoded in the nodes and edges and their interactions.

We consider challenging benchmarks with important edge features to evaluate our approach. We use the term *edge-important graph datasets* to refer to datasets containing graphs where edge features play a crucial role in conveying information. Specifically, we introduce a new synthetic dataset of grid mazes whose graphs are based on Markov decision processes. In this setting, edge attributes encode the probability of moving

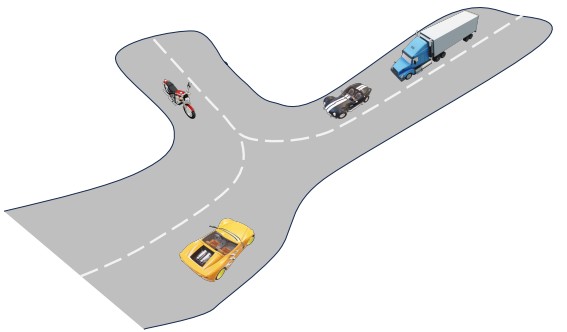

(a) Simulation of a possible traffic scene with two cars, a truck, and a motorcycle.

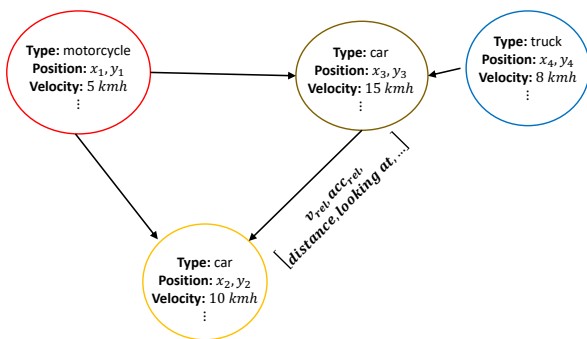

(b) The corresponding graph representation of the scene.

Figure 1: **Example of a graph with attributed edges.** It illustrates a graph in Figure 1b that represents the driving scene in Figure 1a. Each node represents an entity within the scene and encodes node attributes such as one of the types car, truck, motorcycle, velocity, position, history trajectory, etc. Additionally, edges are attributed with relative information such as distance, velocities, relative accelerations, and a flag indicating whether the vehicle's driver, represented by the source node, looks at the vehicle represented by the target node. For illustration purposes, we use a graph with only a few edges, but additional edges could be incorporated to represent more complex relationships.

between grid cells. Additionally, we offer traffic scene generation on nuScenes (Caesar et al., 2020) as a new benchmark for evaluating edge-important generative graph methods. Finally, we define and estimate edge-related error metrics, allowing us to compare edge capabilities of generative models quantitatively. **Our main contributions can be summarized as follows:**

- We extend the graph generation task to enable the generation of more expressive graph structures by formulating the graph generation with edge attributes. We propose a joint SDE framework for generating graphs with this information and demonstrate the importance of generating all graph components encompassing multiple node and edge attributes. We establish new links between graph generation in MDPs and real-world traffic scenarios. We advocate for a comprehensive benchmark for edge-based graph generation and lay the groundwork for future research on integrating multiple edge attributes into various graph-based applications.

- We introduce a novel inductive bias for score-based models in graph generation, leveraging a newly formulated SDE approach that captures the interplay between edges and nodes. Additionally, our model incorporates an architectural bias to facilitate the propagation of edge information for better score estimation.

- We thoroughly evaluate our approach on diverse benchmarks and conduct ablation studies. Our results demonstrate superior performance over baseline models, excelling across various standard evaluation protocols for graph generation tasks, particularly in edge metrics.

## 2 Related Work

**Score-based generative models.** Diffusion and score-based models represent generative models that sample new data by iteratively denoising a simple, pre-defined distribution (Sohl-Dickstein et al., 2015; Song & Ermon, 2019; Ho et al., 2020). Song et al. (2021) showed that these methods can be described in a unified framework of SDEs. Thus, we will use the terms diffusion and score-based models interchangeably. The diffusion process consists of the forward pass, where noise is gradually added to the data until it converges to a normal Gaussian distribution, and the reverse pass, where a backward SDE is integrated using a denoising model. New samples are defined as the convergence points of the reverse pass. While several graph generative frameworks exist (Wu et al., 2020; Zhou et al., 2020), we focus on score-based approaches.

**Discrete and continuous graph diffusion models.** Haefeli et al. (2022) suggest discrete perturbations of the data distribution through a denoising diffusion kernel. Similarly, DiGress (Vignac et al., 2023) uses discrete diffusion methods (Austin et al., 2021) to produce discrete graphs containing categorical node and edge values. Recently, GraphARM (Kong et al., 2023) designed a node-absorbing autoregressive diffusion for efficient and high-quality sampling. Others proposed a discrete diffusion process that utilize graph sparsity to gain efficiency (Chen et al., 2023). While it is argued that discrete modeling of graphs may be beneficial, it is unclear how to sample real-valued attributes in existing frameworks. We insist that many real-world problems are naturally defined by continuous values, which requires the development of a general graph generative model. To this end, several works have proposed score-based methods for graph generation, though they often face limitations in modeling edge attributes. Niu et al. (2020) introduced a permutation-invariant model based on graph neural networks (GNNs) for learning data distributions of adjacency matrices. To extract binary neighborhood information, the real-valued diffusion output is discretized via thresholding. Subsequently, GDSS (Jo et al., 2022) uses separate stochastic differential equations to model node attributes and the adjacency matrix independently. Yet, this separation may limit the information exchange between nodes and adjacency. Recently, SwinGNN (Yan et al., 2023) proposed a non-invariant approach that permutes the adjacency matrix for graph generation. While this method addresses the permutation invariance, it still fails to generate edge features, which remain unmodeled in this framework.

Following the success of graph score-based models, we are motivated to further extend this framework to include edge features. This is achieved by a careful inspection of GDSS (Jo et al., 2022), which results in the conclusion that separate scores for different components may lack context for the generation task. While trivial extensions exist, we find them to be unsatisfactory in solving even simple edge feature generation tasks, let alone challenging graph benchmarks. Instead, we address this problem by proposing a joint SDE for all graph components, combined with a dedicated GNN architecture to exploit edge features. We show that our method greatly outperforms naive adaptations, demonstrating the necessity of each and every component we introduce as a whole.

**Edge-based GNNs.** We also mention a few works that consider edge-based GNNs for various tasks. Schlichtkrull et al. (2018) offered a decomposition for relational data. In Gong & Cheng (2019), the authors exploit edge features via a doubly stochastic normalization. Similarly, Wang et al. (2021) extended GNNs to handle edge features and node features. Motivated by these works, we explore graph generation by considering node and edge attributes.

## 3 Background

We briefly discuss the essential components of score-based models on Euclidean domains $\mathbb{R}^d$. We refer to (Song et al., 2021) for further details. A *diffusion process* is defined by $\{x(t)\}_{t=0}^T$ with $t \in [0, T]$, where $x(0)$ is sampled from the data distribution $x(0) \sim p_0$; and $x(T) \sim p_T$, with $p_T$ being a simple prior distribution such as standard normal. Diffusion processes are the solutions of SDEs of the form,

$$dx = f(x, t)dt + gdw \ , \tag{1}$$

where $f(\cdot, t) : \mathbb{R}^d \to \mathbb{R}^d$ is the drift coefficient, $g \in \mathbb{R}$ is the diffusion scalar, and w is a standard Wiener process. We adhere to standard notations and denote the probability density of $x(t)$ as $p_t(x)$, and the transition kernel from $x(s)$ to $x(t)$ for $s < t$ as $p_{st}(x(t) \,|\, x(s))$.

The process described in Eq. 1 is generative, as it allows for the generation of samples from $x(T) \sim p_T$, which can then be propagated backward through a reverse process. A well-known result by Anderson (1982) shows that the following reverse-time SDE is the reverse diffusion process,

$$dx = [f(x, t) - g^2 \nabla_x \log p_t(x)]d\bar{t} + gd\bar{w} \ , \tag{2}$$

where $\bar{w}$ is a reverse-time Wiener process, and $d\bar{t}$ denotes an infinitesimal negative timestep. Integration of Eq. 2 from time $T$ to time 0 allows an effective sampling from $p_0$. Unfortunately, estimating the score, $\nabla_x \log p_t(x)$, is difficult for all timesteps except for $t = T$, which is defined as the prior distribution. Thus,

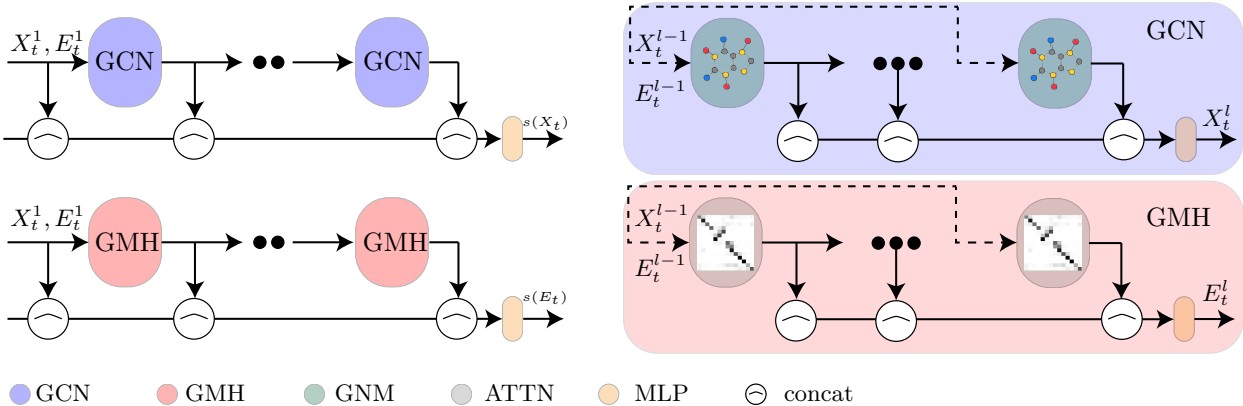

Figure 2: **Our architecture**. On the left, we show the score modules (Eq. 15). On the right (Blue), our GCN module is constructed of GNM (Eq. 9) layers. On the right (Red), the GMH module is constructed of attention (ATTN) layers.

score-based models (Song & Ermon, 2019) train an estimator $s_\theta(\mathrm{x}, t)$ with the objective of

$$\min_\theta \mathbb{E}_t \{\mathbb{E}_{x_0, \mathrm{x}_t} \left[ |s_\theta(\mathrm{x}_t, t) - \nabla_{\mathrm{x}_t} \log p_{0t}(\mathrm{x}_t \,|\, \mathrm{x}_0)|_2^2 \right] \} \ . \tag{3}$$

## 4 Method

Our method for generative modeling of graphs is based on two novelties. First, we propose a joint score-based model of node and edge attributes (Sec. 4.1). The sample's score is evaluated for all graph components jointly. Second, we combine node, edge, and adjacency information in the attention module (Sec. 4.2). With these two key ingredients, we achieve a modeling of graphs as a whole.

### 4.1 A Joint SDE Model

The main goal of our model is to represent the data distribution of graphs, denoted $p_0$. A graph with $n$ nodes is a 2-tuple $G = (X, E)$, where $X \in \mathbb{R}^{n \times u}$ are the node attributes and $E \in \mathbb{R}^{n \times n \times v}$ is the edge attributes tensor. We refer to App. B.1 for a detailed formulation. Importantly, the adjacency matrix $A \in \{0, 1\}^{n \times n}$ can be recovered from $E$ by a simple mask such as

$$A := \sigma(\max_k |E_{ijk}|) \ , \tag{4}$$

where $\sigma = 0$ if $\sigma(x) < \epsilon$ and else $\sigma = 1$. The motivation behind the masking is to emulate the graph structure throughout the process and at the conclusion of the GNN feedforward operation. We chose $\epsilon = 0.01$ for all datasets. Thus, encoding $G = (X, E)$ is sufficient to fully capture the underlying structure of the graph. The choice of Eq. 4 is particularly suitable for the datasets considered in our study. As we show hereunder, it effectively filters out low-probability transitions for MDPs, while for traffic generation, small distances correspond to invalid links. In both cases, $\epsilon$ serves to remove insignificant connections, improving the model's focus on meaningful relationships.

We would like to generate new graphs $G \sim p_0$, which we achieve by defining a diffusion process from $p_0$ to $p_T$ (and back), as we describe below. We follow the general outline in Sec. 3. A diffusion process on $\{G_t = (X_t, E_t)\}_{t=0}^T$ is given by the SDE,

$$\mathrm{d}G_t = f(G_t, t)\mathrm{d}t + g\mathrm{dw} \ , \tag{5}$$

where $f$ is the drift transformation on a set of graphs $\mathcal{G}$, i.e., $f(\cdot, t) : \mathcal{G} \to \mathcal{G}$. The corresponding reverse-time SDE reads

$$\mathrm{d}G_t = [f(G_t, t) - g^2 \nabla_{G_t} \log p_t(G_t)]\mathrm{d}\bar{t} + g\mathrm{d}\bar{\mathrm{w}} \ , \tag{6}$$

where in Eq. 5 and Eq. 6, we abuse the notation that appeared in Eq. 1 and Eq. 2, while keeping the equivalent meaning for $g$, w, $\bar{\text{w}}$ and d$\bar{t}$ on graphs. Analogously to Eq. 3, the score $\nabla_{G_t} \log p_t(G_t)$ is estimated using a graph neural network whose objective is

$$\min_{\theta} \mathbb{E}_t \{ \mathbb{E}_{p_0, p_{0t}(G_t|G_0)} \left[ |s_\theta(G_t, t) - \nabla_{G_t} \log p_{0t}(G_t \,|\, G_0)|_2^2 \right] \} \,, \tag{7}$$

where $G_0 \sim p_0$, $G_t \sim p_{0t}(G_t \,|\, G_0)$.

GDSS considered a diffusion process similar to Eq. 5. However, they use separate processes for the nodes and adjacency instead of solving the joint SDE. Consequently, nodes and neighbors affect each other only through the score function. On the other hand, we aim to solve it jointly for nodes and edges, allowing them to interact during the diffusion process through the score calculation. This modification will also be emphasized below, where we elaborate on our graph neural network.

## 4.2 Node and Edge-Dependent GNN

Similar to existing works (Niu et al., 2020; Jo et al., 2022), we adopt the framework of Graph Neural Networks (Wu et al., 2020). This architecture maintains permutation equivariance, ensuring that the model learns a desired permutation-invariant distribution (Niu et al., 2020). Moreover, we utilize the graph multi-head attention module (Baek et al., 2021). A fundamental element within our strategy is the Graph Neural Module (GNM), where nodes, edges, and adjacency exchange information. The illustration of our architecture is given in Fig. 2.

**Graph neural module.** Given an intermediate estimation of node and edge attributes, denoted by $X_t$ and $E_t$, respectively, the GNM module is defined via

$$\text{GNM}(X_t, E_t) := \bar{A}_t X_t W_X + \tanh(B[\text{rep}(\bar{A}_t) \odot E_t W_E]) \,, \tag{8}$$

where $\odot$ is the element-wise product, $W_X, W_E$ are neural network weights. $B[\cdot]$ sums the values of each node incoming edges feature-wise and the operator $\text{rep}(\cdot)$ takes a matrix and repeats it $v$ times along the third dimension. Inspired by Kipf & Welling (2017), we construct the matrix $\bar{A}_t$ by scaling the adjacency $A_t$ with the degree matrix $D_t$, i.e., $\bar{A}_t = D_t^{-\frac{1}{2}} \odot A_t \odot D_t^{-\frac{1}{2}}$, where $D_t$ is a diagonal matrix encoding the number of edges per node. Finally, $A_t$ is extracted via Eq. 4. The GNM module learns how to propagate information to each node from its neighboring nodes while also absorbing information from incoming and outgoing edges. As shown in a later ablation experiment, this capability enables the propagation of edge feature information. Essentially, the left side of the addition learns to propagate node features to other nodes and edges, and the right side of the addition does so for the edge features. Finally, the output shape of this operation is $b \times n \times d$ where $b$ is the batch size, $n$ is the number of nodes in the graph and $d$ is the number of features.

**Attention module.** We also employ a commonly-used attention module, ATTN (Baek et al., 2021), $\text{ATTN}(X_t, E_t) := \text{avg}\left(Q_t K_t^T / \sqrt{d_t}\right)$, with $Q_t := \text{GNM}_Q(X_t, E_t)$ and $K_t := \text{GNM}_K(X_t, E_t)$, $\text{avg}(\cdot)$ is the mean over the axis of the different attention channels , and $d_t$ is the attention dimension. Previous studies, such as Jo et al. (2022), have demonstrated the effectiveness of attention as a simple yet powerful model. It facilitates efficient information propagation through both nodes and edges. The output dimension of the attention operation is $b \times n \times n \times k$ where $b$ is the batch size, $n$ is the nodes size and, $k$ is the number of features on each edge.

**Our graph neural network.** We utilize the GNM and ATTN modules to construct our full graph neural network to compute the score $s_\theta(G_t, t)$. To simplify notation, we define $H(\{h_j\}, J, M)$ as the module that takes a collection of vectors $\{h_j\}$ with $J$ elements, concatenates them, and feeds the result through a multilayer perceptron (MLP) $M$:

$$H(\{h_j\}, J, M) := M\left(\text{concat } [h_j]_{j=1}^J\right) \,. \tag{9}$$

$J$ is determined by a hyper-parameter and plays a role in enabling the model to capture multiple propagation flows at each level of the graph neural network. Then, we define two components that will be used to generate

the node and edge attributes. The graph convolutional network (GCN) and graph multi-head attention (GMH) are given by

$$\text{GCN}(X_t, E_t) = H(\{\text{GNM}(X_t, E_t)_j\}, J, M_\varphi) \,, \tag{10}$$

$$\text{GMH}(X_t, E_t) := H(\{\text{ATTN}(X_t, E_t)_j\}, J, M_\phi) \,. \tag{11}$$

The GCN and GMH components compress information across $J$ different activations.

Finally, we define the score $s_\theta(G_t, t)$ by

$$s_\theta(G_t, t) := (s_\theta^X(X_t, E_t, t), s_\theta^E(X_t, E_t, t)) \,. \tag{12}$$

To estimate the aforementioned score function, we use feed-forward neural networks $F_X$ and $F_E$ as follows. Given initial node and edge attributes, denoted as $X_t^1 = X_t$ and $E_t^1 = E_t$ respectively, the model sequentially alters its inputs as they pass through the layers by

$$X_t^l := F_X^l(X_t^{l-1}, E_t^{l-1}) \equiv \text{GCN}(X_t^{l-1}, E_t^{l-1}) \,, \tag{13}$$

$$E_t^l := F_E^l(X_t^{l-1}, E_t^{l-1}) \equiv \text{GMH}(X_t^{l-1}, E_t^{l-1}) \,. \tag{14}$$

Here, $X_t^l, E_t^l$ denote the node and edge attributes representing the output of the $l$-th layer, $l \in [1, L]$. It incorporates multiple hierarchical latent representations, enabling the model to capture multiple propagation steps and different levels of information abstraction. We also considered using GMH instead of GCN to compute the node scores; however, we found that both approaches yielded similar performance. Consequently, we opted for GCN due to its simplicity. Conversely, we found that utilizing GMH was crucial for achieving superior results for calculating edge scores. Our computation is completed by

$$s_\theta^X(X_t, E_t, t) = H(\{X_t^l\}, L, M_{\theta_X}) \,, s_\theta^E(X_t, E_t, t) = H(\{E_t^l\}, L, M_{\theta_E}) \,. \tag{15}$$

Importantly, our Eq. 8 allows for a proper digestion of the edge information by the score network and its propagation through the entire score model. We consider this formulation to be crucial for estimating the score, as demonstrated in Sec. 5.4. Finally, we briefly conduct a complexity analysis. The attention mechanism governs the time complexity. Specifically, both GMH and GCN modules introduce an $\mathcal{O}(n^2)$ component for time that scales linearly with the number of attention heads and the input/output feature dimensionalities. Storage for edge features requires $\mathcal{O}(n^2 v)$, with $v$ being the number of features. Thus, our approach is similar in time and memory complexity to other state-of-the-art models such as DiGress (Vignac et al., 2023) and GDSS (Jo et al., 2022). In addition to the theoretical analysis, we conduct an empirical complexity evaluation, detailed in (App. C.1).

## 5 Experiments

We tested our qualitative and quantitative methods on diverse real-world and synthetic dataset benchmarks. The objective of the model is to learn from observed graphs the underlying distribution and be able to generate new unseen graphs that follow the same distribution. We refer to App. B.1 for a detailed problem specification. The particular objectives of this study are:

- As our main focus is edge features generation, we show in Sec. 5.1 that incremental modifications of GDSS are ineffective, highlighting the non-trivial importance of our approach.

- We introduce a new challenging synthetic dataset of Markov decision processes (Sec. 5.2). Further, we present a new use case: a real-world traffic generation task (Sec. 5.3). To the best of our knowledge, we are the first to tackle this task via graph generation.

- We evaluate the score estimation quality (Sec. 5.4), and we conclude by ablating our model to analyze the contribution of every component to its performance (Sec. 5.5).

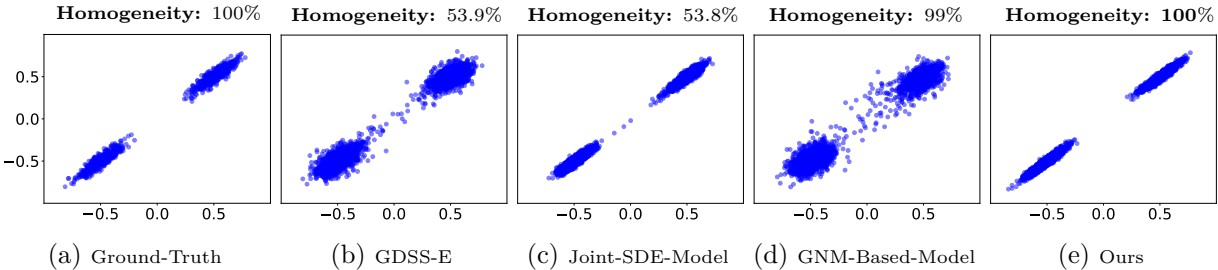

Figure 3: Our ablation study shows that GDSS-E and the other variants yield inferior distributions compared to our approach concerning the ground-truth data distribution estimation.

## 5.1 Synthetic Dataset Ablation Experiment

In this section, we conduct an in-depth study to assess the performance gains from incorporating our different components. Our goal is to demonstrate that the current baseline, even with incremental adjustments for edge feature generation, is ineffective for this task, whereas our approach effectively learns both inter- and intra-edge feature attributes as intended.

**Dataset.** We utilize a synthetic dataset of 1000 complete graphs with ten nodes each, with only edge attributes. The nodes do not contain any information. Let $E$ be the set of edges in a sampled graph. Each $e \in E$ belongs to $\mathbb{R}^2$ and the two features in $e$ are sampled randomly from *only one* of the Gaussian clusters depicted in Fig. 3a. Additionally, the graph edges are homogeneous, i.e., they are all sampled from the same cluster (the upper right or the lower left).

**Baseline and variants.** To create a solid baseline, we adapt GDSS to handle multiple edge features, and we name it *GDSS-E*. Refer to App. B.4.2 for more details. We consider GDSS-E to be a vanilla model without any of the components that we proposed in this work. Then, to ablate our two model components, we separate each and add them to the vanilla baseline model. We denote the baseline with our joint SDE model (Eq. 7) as *joint-SDE-Model*, and the baseline with our GNM model (Eq. 8) as *GNM-Based-Model*. Finally, our approach is based on GDSS-E and comprises both components. Note that all the variant's score estimations are similar to the base model.

**Evaluation.** Fig. 3 contains scatter plots of the edge distributions for the different baselines. For this visualization, we generate 50 complete graphs with 5000 edges in total. The desired result would be a generation of edge features similar to the ground truth in Fig. 3a. A good model should generate the same visual clustering as the ground truth. Further, to evaluate the edge features quantitatively, we consider the homogeneity of edges. We check the percentage of graphs that are homogeneous, meaning all edges in the graph belong to a single cluster, as in the real data. Then, we compute the average percentage of this test for the 5000 generated samples per method. We detail the homogeneity score above each plot, where good models should yield 100% as the ground truth.

**Results.** Fig. 3b shows that GDSS-E roughly approximates the distribution. However, the two clusters appear blurred, making it difficult to differentiate between them. In addition, it fails to learn the homogeneity characteristics of the data. The joint-SDE-Model (Fig. 3c) presents improved results by estimating denser Gaussian clusters. Alas, it fails to yield a fine-grained generation as some samples are outside the original distribution (e.g., the points around 0). Further, this model also fails in the homogeneity task. The GNM-Based-Model (Fig. 3d) generates blurred and non-separated clusters. Nevertheless, it successfully models homogeneous edge features, achieving a 99% homogeneity score. Finally, our approach (Fig. 3e) demonstrates a similar distribution to the ground truth in terms of separation and clusters' structure, as well as a perfect homogeneity score.

These results shed light on the importance of our components. On the one hand, our joint SDE process accurately models the underlying distribution, but it struggles with preserving homogeneity, i.e., with interactions between the graph edges. On the other hand, the GNM-based model succeeds in maintaining

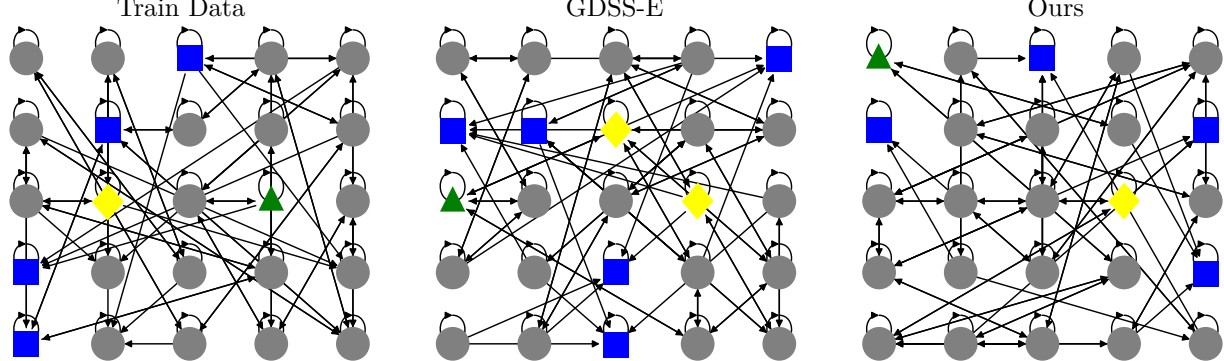

Figure 4: A qualitative comparison between the original data (left), GDSS-E (middle), and our method (right) on the deterministic MDP grid maze dataset. Blocks are colored in blue (squares), and start and finish nodes in yellow (diamonds) and green (triangles), respectively. Our graphs consistently have four blocks, one start node, and one finish node, as required.

homogeneous aspects, but it is challenged by the data distribution. We conclude that our new components are important: 1) the joint SDE process is crucial in modeling intra-edge interactions, and 2) the GNM module is instrumental in capturing inter-edge relationships.

## 5.2 Generative MDPs

Reinforcement Learning (RL) environments typically consist of multiple states and the probabilities to move from one state to another. These environments are often formalized as Markov Decision Processes (MDPs), which can be viewed as directed graphs, where nodes represent states and edge attributes represent the transition probabilities. We strive to explore the connection between generative modeling of graphs and MDPs. Indeed, access to many diverse RL environments is often limited in practice, and we aim to extend and diversify available environments.

In this context, we introduce a new synthetic MDP dataset of grid mazes. Each grid has $5 \times 5$ cells (nodes), including a start cell and a finish cell and 25 cells in total. There are also block cells that the agent cannot traverse, while the remaining cells are empty and walkable. The agents take one action per cell {up, left, down, right}. The maze is encoded via a graph whose nodes are the cells, and edges are the optional transitions. The nodes in the graph, each with a single feature, represent the discrete category of each cell {start, finish, block, empty}. The attributes on the edges are the continuous probabilities of moving from the current cell to one of its adjacent cells. Here, we consider two settings of using this data: (1) a deterministic grid maze, where edge features are binary in $\{0, 1\}$; and (2) a non-deterministic grid maze, where edge attributes are probabilities in $[0, 1]$, and the sum of all features per cell is one. Further details on these datasets and their MDP graphs are provided in App. B.2.1.

A unique characteristic of our MDP graphs is their multiple attributes per edge. To the best of our knowledge, this scenario has yet to be studied in existing generative works. In particular, prior works are designed

Table 1: Quantitative graph generation metrics on deterministic and non-deterministic grid mazes.

| Method | deterministic | | | | | | | | | non-deterministic | | | | | | | | | |
|---|---|---|---|---|---|---|---|---|---|---|---|---|---|---|---|---|---|---|---|
| | deg↓ | cl↓ | un↑ | no↑ | MV↑ | VS↑ | B↓ | SF↓ | E↓ | deg↓ | cl↓ | un↑ | no↑ | MV↑ | MDV↑ | VS↑ | B↓ | SF↓ | E↓ |
| GDSS-E | 0.73 | 0.06 | 97 | 100 | 34% | 9% | 0.96 | 1.28 | 2.23 | 0.40 | 0.02 | 99 | 100 | 6% | 1% | 26% | 0.39 | **0.83** | **0.4** |
| SwinGNN-E | 0.44 | 0.064 | 100 | 100 | 17% | **62%** | 2 | 1.68 | 3.69 | 0.51 | 0.05 | 100 | 100 | 21% | 1.9% | **72%** | 2.8 | 2.6 | 5.4 |
| Ours | **0.17** | **0.006** | 100 | 100 | **68%** | 34% | **0.1** | **0.58** | **0.48** | **0.31** | **0.013** | 100 | 100 | **38%** | **6%** | 33% | **0.02** | 0.88 | 0.8 |

to construct only a single edge value. To compare our approach against strong baselines, we consider the state-of-the-art GDSS and modify it to GDSS-E as discussed in Sec. 5.1. Furthermore, we use a variant of SwinGNN (Yan et al., 2023), a state-of-the-art score-based graph generative model, to generate multiple edge and node features as an additional baseline and denote it as SwinGNN-E. We do not consider DiGress as a baseline due to significant disparities in the output format. DiGress (Vignac et al., 2023) solely produces discrete attributes, whereas our requirements necessitate continuous ones. Moreover, adapting DiGress to generate multiple edge attributes entails non-trivial modifications, rendering it unsuitable for direct comparison in our experimental framework.

To quantitatively evaluate the graphs, we utilize common metrics such as the degree (deg) and cluster (cl). We do not use the orbit metric since the grid maze MDPs are directed cyclic graphs. Additionally, we adapted the uniqueness (un) and novelty (no) metrics (Martinkus et al., 2022) to evaluate the models' ability to generate graphs that differ from the training set and are distinct from each other. Further, we introduce five new dataset-specific and edge-based metrics that measure the quality of generated graphs and edge features. Valid solution (VS) tests if the grid is valid, i.e., it has start and finish cells with a viable route between them. Blocks (B) measures the distance between the average number of blocks in the grid, where in our dataset, the ground-truth value is four. Start and finish (SF) calculates the distance between the average number of start and finish cells. To clarify, in our context, start and finish are nodes classified as yellow or green, as shown in Fig. 4. There is exactly one starting cell and one ending cell. Empty (E) computes the distance between the average number of regular cells, which is always 19. For (B), (SF), and (E), the distance is defined as the absolute difference between the fixed original number of cells and the corresponding values in the generated graph. For example, if a graph is generated with one starting cell, one finishing cell, and five blocks, then the calculations would be $F = |2 - 2| = 0$, $B = |5 - 4| = 1$, and $E = |19 - 18| = 1$. The final result is the average of the same calculation for each graph. Finally, MDP validity (MV) estimates the percentage of valid edge features in the generated graphs. Features are valid if the sum of the outgoing edges of a node is equal to 1. The results of our evaluation on grid maze MDPs are shown in Tab. 1.

**The deterministic setting.** Our approach better captures the graph statistics measured by the degree (deg) and cluster (cl) metrics, showing a significant gap concerning GDSS-E and SwinGNN-E. Further, our graphs' edge-based metric, MV, is twice the baseline result, i.e., **68**% vs. 34%. These results emphasize our model's ability to capture edge attribute complexities. Finally, our model achieves strong results in the metrics that estimate node generation compared to GDSS-E. It is essential to highlight that although SwinGNN achieves a high percentage of valid solutions (VS), this metric does not consider whether there are multiple start or finish points. While its B, SF, and E metrics are relatively high, indicating that the model struggles to grasp the statistical features of the graph nodes as required compared to our model. We also present a qualitative comparison of the real training data, the generated graphs obtained with GDSS-E, and our approach. Fig. 4 shows a sample from the training data (left), a graph generated with GDSS-E (middle), and our generated graph (right). The colored nodes are blocks (blue) and start and finish cells (yellow and green, respectively). Our method yields a valid graph, respecting the correct number of blocks and start and finish points. In contrast, GDSS-E has two starting points and five block nodes.

**The non-deterministic case.** In this setting, where edge features are real-valued, edge features are valid if their MV measure is $\epsilon = 0.001$ close to one. Further, the edge values follow a specific pre-defined distribution, and thus, in addition to MV, we also measure the MDP distribution validity (MDV). Namely, the percentage of edges that follow the distribution above. Notably, the measures in Tab. 1 show that our model performs $\approx$ **3** times better than the baseline on the edge-related metrics, MV and MDV. These results affirm our framework's ability to capture and generate diverse and complex multiple-edge attributes effectively.

### 5.3 nuScenes: traffic scene generation

Learning traffic scenes can hugely contribute to autonomous driving. To the best of our knowledge, we are the first to suggest a generation of traffic scenes as graphs. We leverage the idea that a scene with elements such as cars, tracks, traffic lights, lanes, and

Table 2: Quantitative metrics on nuScenes.

| Method | deg↓ | cl↓ | un↑ | no↑ | V↓ | O↓ | L↓ | CR↓ | LA↓ |
|--------|------|-----|-----|-----|-----|-----|-----|-----|-----|
| GDSS-E | 1.05 | 0.03 | 15 | 34 | 3.9 | **0.66** | 0.96 | 0.5% | 208 |
| SG-E | 0.79 | 0.01 | 39 | 42 | 0.76 | 1.27 | 1.08 | 0.6% | 302 |
| Our | **0.77** | $6e^{-7}$ | **51** | **51** | **0.36** | 0.8 | **0.08** | **0.3%** | **194** |

more can be represented as a graph. In this graph,
each node is an agent containing the trajectory across time, and each directed edge represents the effect of one agent on another. Edge features such as Euclidean distances and angles are optional. In what follows, we use nuScenes (Caesar et al., 2020), a public dataset that is broadly used for trajectory prediction (Liu et al., 2021). Primarily, the challenge is to predict a future trajectory from the history traces of a road participant. We transform each scene into a vector-graph representation, similar to VectorNet (Gao et al., 2020). The latter work was the first to utilize a graph representation of the scenes, where nodes represent agents and map elements, which are later processed via GNNs to predict the target. Refer to App. B.2.2 for more details.

We evaluate our model using standard graph metrics for directed graphs alongside established traffic generation protocols (Tan et al., 2023; Feng et al., 2023). Specifically, we compute Maximum Mean Discrepancy (MMD) (Gretton et al., 2012) metrics for vehicles (V), objects (O), and lanes (L), as well as collision rate (CR) and lane alignment (LA). In particular, we assess MMD over the $x$ and $y$ coordinates of vehicle trajectories (V), lane curves (L), and other map objects (O). Collision Rate (CR) quantifies the frequency of collisions between generated agents, while Lane Alignment (LA) measures the distance from each agent's trajectory to the nearest lane, reflecting the tendency of agents to follow lane centerlines—a natural road behavior. For more details on these metrics and evaluation protocols, refer to (Tan et al., 2021). Further information on graph representation, training, and evaluation protocols is provided in App. B.2.1. Tab. 2 presents our results compared to the GDSS-E and SwinGNN-E (SG-E) baselines, demonstrating that our model outperforms these baselines on most general and traffic-specific metrics. Notably, our model captures lane location statistics (L) with approximately a tenfold improvement over the baselines.

## 5.4 Comparison of Score Losses

In this experiment, we compare the behavior of graph generation models in terms of the node and edge score losses on the train and test sets. We report these losses throughout training on the nuScenes dataset in Fig. 5. The left column corresponds to the train set, and the right to the test set. The top row shows the node losses, whereas the bottom row shows the edge losses. We use blue and orange for the loss measures of our method and GDSS-E, respectively. While the node score losses are comparable for both models, our method yields significantly better edge losses.

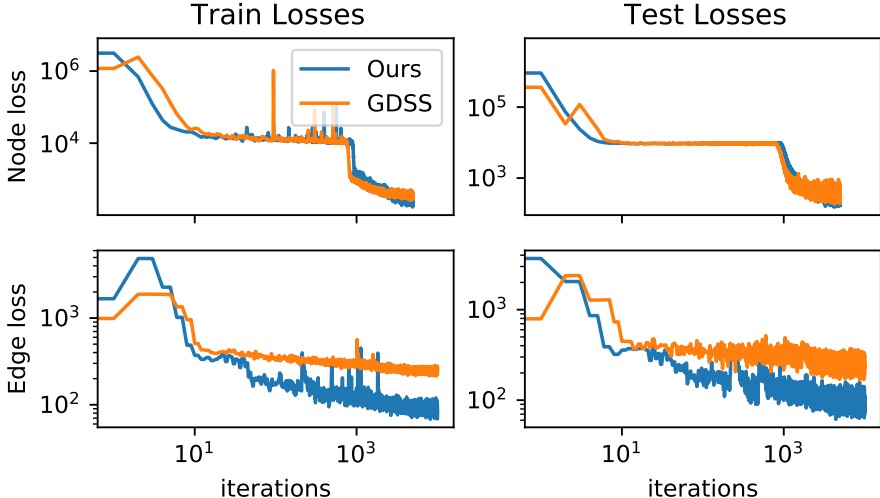

Figure 5: We plot the node and edge scores of GDSS-E and our model on the train (left) and test (right) sets.

## 5.5 Ablation Study

**Edge-based graphs benchmark ablation.** We extend our ablation study in Sec. 5.1 and consider the four model variants. We perform our quantitative ablation over deterministic MDP (MDP-D), non-deterministic MDP (MDP-ND), and nuScenes. We report the results of the metrics deg, cl, and MV or LA in Tab. 3. We

Table 3: Ablation study of the four variants of our model.

| Method | Planar | | | SBM | | | MDP-D | | | MDP-ND | | | nuScense | | |
|---|---|---|---|---|---|---|---|---|---|---|---|---|---|---|---|
| | deg↓ | cl↓ | orb↓ | deg↓ | cl↓ | orb↓ | deg↓ | cl↓ | MV↑ | deg↓ | cl↓ | MV↑ | deg↓ | cl↓ | LA↓ |
| GDSS-E | 0.945 | 0.96 | 0.66 | 0.74 | 1.57 | 0.25 | 0.73 | 0.06 | 34% | 0.40 | 0.02 | 26% | 1.05 | 0.03 | 208 |
| Joint-SDE-Model | 1.02 | 0.94 | 0.26 | **0.2** | 1.04 | 0.05 | 0.71 | 0.05 | 57% | 1.67 | 0.54 | 2% | 1.4 | 0.02 | 243 |
| GNM-Based-Model | 0.038 | 0.95 | 0.22 | 0.34 | 0.7 | 0.05 | 0.23 | 0.02 | 55% | 0.35 | 0.07 | 33% | 0.99 | $1e^{-5}$ | **179** |
| Ours | **0.025** | **0.38** | **0.23** | 0.46 | **0.63** | **0.04** | **0.17** | **0.006** | **68%** | **0.31** | **0.013** | **33%** | **0.77** | $6e^{-7}$ | 194 |

omit some metrics due to space constraints. However, the trend is similar in those metrics as well. Our results indicate that the proposed model obtains the best results across all datasets and metrics, except for LA in nuScenes. Further, we find that only incorporating edge-based GNM, leads to inconsistent behavior. However, jointly modeling node and edge attributes attains a notable gain in error metrics. Finally, using both components leads to the best results.

**General graphs benchmark ablation.** Although our study focuses on edge-important graph benchmarks, we apply our method to general graph generation tasks and extend the ablation study to show the robustness of our model to different diverse datasets. To leverage the edge attribute abilities of our model, we augment every graph with edge attributes per edge. Specifically, we compute the $n$-th power of the adjacency matrix, and then, for each edge $e_{ab}$ between nodes $a$ and $b$, we assign the corresponding value encoded in the power matrix. The edge features contain the number of paths between $a$ and $b$ with $n$ steps, where we set $n = 2$. In Tab. 3, we report the results on the Planar and SBM datasets. We observe a trend similar to the previous ablation, and our method outperforms the other variants. In addition to the ablation study, we compare general graph benchmarks with augmented features at App. C.3. Furthermore, we test our model on a real-world molecule dataset at App. C.3.2. In both experiments, we show that our model can learn the graph distribution better with edge features, and we achieve competitive results concerning solid baselines.

# 6 Conclusion

While graph generation models must consider all graph elements and their interactions, existing works focus only on adjacency and node attributes. Further, score-based methods utilize a separate diffusion process per graph element, which limits the interaction between the sampled components. This work suggests a joint score-based model for node and edge features. Our framework maximizes learning from graph elements by combining node and edge attributes in the attention-building block module. Moreover, node, edge, and adjacency information are mutually dependent by construction in the diffusion process. We extensively test our approach on multiple synthetic and real-world benchmark datasets compared to recent strong baselines. Further, we introduced a new synthetic dataset for benchmarking edge-based approaches. Our results show that exploiting edge information is instrumental to performance in general and in edge-related metrics.

In the future, we aim to incorporate certain inductive biases into the generation pipeline. For instance, challenging benchmarks such as MDPs and nuScenes could greatly benefit from this approach, as current methods are limited in their ability to fully capture the underlying rules of the data. Additionally, generating new samples with diffusion frameworks is costly and difficult to scale to large graphs. We plan to address these limitations by enabling variation in the number of nodes within the diffusion process, thus allowing for non-fixed-size graphs. Finally, increasing the expressivity of GNNs by relaxing the permutation invariance property represents an exciting avenue for further research.

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

## A Appendix

## B Additional Details

### B.1 Problem Statement

The main problem our work focuses on is given a group of observed graphs drawn from i.i.d unknown distribution denoted $p_0$. We want to learn a model $M$ that will allow us to sample a new unseen graph $G = (X, E)$ where $X \in \mathbb{R}^{n \times u}$ represents the node attributes and $E \in \mathbb{R}^{n \times n \times v}$ represents the edge attributes tensor from the observed distribution $p_0$. This problem is frequently called "unconditional generation" (Guo & Zhao, 2022). While prior works focused on sampling graphs from $p_0$ that contain only node features, that are graphs with limited expressiveness in the sense that edge information depicts solely the graph topology, sampling high-dimensional edge features is somewhat overlooked, disabling the ability to learn distributions $p_0$ that contain edge features. Our work focuses on learning a model $M$ that will learn to generate new samples from observed $p_0$ containing node and edge features.

### B.2 Datasets

#### B.2.1 MDP Grid Maze - Datasets

**Motivation.** We propose an innovative link between graph generation techniques and Markov Decision Processes (MDPs). Generating various environments for agents is crucial in Reinforcement Learning (RL) for effective task learning. However, there are instances where access to diverse environments is limited. Environments can be formalized as MDPs, and MDPs can be represented as directed graphs. Thus, we are motivated to create a new dataset whose graphs contain node and edge attributes and are directed graphs. Such data will diversify the common standard benchmarks today that include undirected graphs and contain only one type of feature, either for nodes or edges.

**Dataset Description.** We create two variants of the MDP grid maze dataset: deterministic and non-deterministic. In both settings, the grid is the same. However, the probability of an action is different. In both datasets, the graph contains 25 nodes. A node $u \in \mathbb{R}^3$ in the graph contains in its first coordinate one of the next possible cell values: $-1$ for the block cell, $0$ for the empty cell, $1$ for the stating cell and finally, $a \sim [0.5, 1]$ for the finish point. Blocks and the finish line could also be considered as prizes or punishments; however, in our graph representation, blocks are cells that are out of reach. The other two coordinates of $u$ represent the $x$ and $y$ positions in the maze. An outgoing edge $e_{uv} \in \mathbb{R}^4$ between node $u$ and $v$ equals to $p(v|u, a)$, which is the probability of getting to node $v$ from $u$ given an action $a$: . There are four actions: $\mathcal{A} = \{left, up, right, down\}$. Note, a valid MDP is where for all state $u \in \mathcal{S}$, $v \in \mathcal{S}$ and actions $a \in \mathcal{A}$ the sum of all actions:

$$\sum_{\substack{a \in \mathcal{A} \\ v \in \mathcal{S}}} p(v|u, a) = 1. \tag{16}$$

An equivalent constraint is that all node's outgoing edges $u$ will sum to 1. Finally, the graph's connectivity is decided by the following rules: (1) Moving toward a cell categorized as block is impossible. Therefore, there is an edge toward blocks with probability zero. (all $e_{uv}$ values are 0). (2) Block cells have no outgoing edges. (3) The grid perimeter is closed; thus, moving outside the grid is impossible. (4) Finally, all other moves are legal. The values of the edge features are determined by the defined probabilities of $p(v|u, a)$, which we will explain later for both deterministic and non-deterministic setups.

In Fig.6, we present the grid (left) and its graph representation (center) without edge values for simplicity (we later present a more straightforward graph with edge values). In addition, we show a permuted representation of the same graph (right). There are 25! ways to represent the same graph. Although it looks completely different, both representations represent the same grid maze. The yellow cell is the starting point, the green cell is the ending point, dark blue is the block cells, and the gray cells are neutral.

In Fig.7, we show the complete representation, including the edges. The figure illustrates an arbitrary MDP graph, where each edge feature serves as a channel. We showcase the four edge feature channels along with their corresponding values. For simplicity, we divide the representation into four different graph representations. In practice, each edge in the data represents the probability per action in an arbitrary coordinate order. Therefore, we have only one graph with multiple edge features. For instance, the inner edge of the top-left yellow node on Fig.7, denoted node "1", would have the value of

$$e_{1,1} = \{1, 0, 1, 0\}, \tag{17}$$

which is the value of this edge given the left, right, up, and down actions in this order. The figure's nodes are distinguished by color and shape for clarity. Blocks are represented as dark blue squares, while start and finish nodes are marked in yellow diamond and green triangles, respectively. Empty walkable nodes are displayed in light blue.

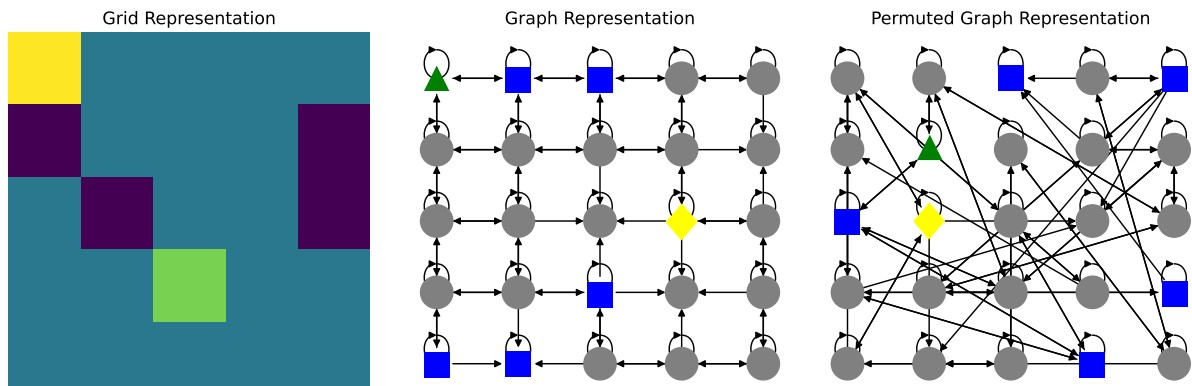

Figure 6: On the left is a grid representation of the MDP grid maze. The yellow cell is the starting point, and the ending is the green cell. The dark blue cells are blocks. In the center, a structured graph represents the grid by the rules described in the appendix. On the right, the same grid is represented by a different permutation of the nodes. Both graphs are equal, and there are $n!$ different graphs, where $n$ is the number of nodes.

**Non-deterministic edges.** MDPs are sometimes non-deterministic. That means, given a state and a desired action, it is only sometimes guaranteed to succeed. We create the non-deterministic dataset variation to simulate this setup and challenge the edge attributes generation. In this dataset, the grids and the nodes are staying the same. However, the edge attributes are now continuous instead of being binary. Still, the outgoing sum of edges from a certain node must sum up to one to be a valid MDP. Further, we decided to apply the next arbitrary distribution over the edge. Denote $|e_{out}^u| = z^u$ as the number of outgoing edges. given a desired action $a$ and nodes $u$, $v$ the rate of success is: $p(v|u, a) = 1 - 0.1 \cdot z^u$. And $p(k|u, a) = 0.1$ for any other node $k$ neighbor of $u$.

**Dataset generator.** We will provide the complete code for generating the grids and their corresponding MDPs. The generator enables control of grid size, number of ending points, and number of blocks. In addition, if these parameters are valid, the code generates only valid graphs with a start and an end. Finally, the generator gets the desired grids and randomly samples grids with the above parameters.

**Dataset statistics.** First, our grids are $5 \times 5$. Second, there is only one ending point. Therefore, we have **one** starting cell and **one** ending cell. Finally, we set the number of blocks to be precisely **4**. These configurations are for every grid in the dataset. We generate 1000 valid grids and split them into 80% training and 20% for testing and validations.

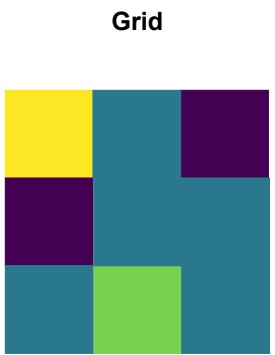

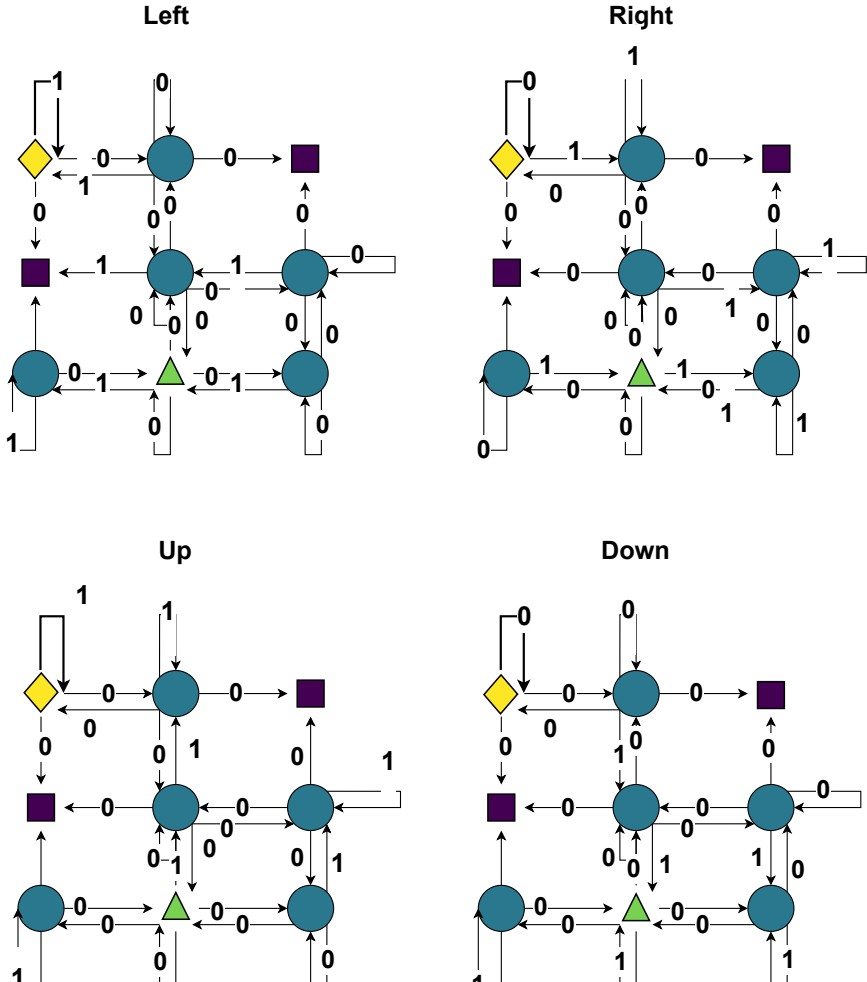

Figure 7: In the first line, a $3 \times 3$ grid example. Note that in the real dataset, the grids are $5 \times 5$. Below, we present the four actions and show the probabilities of moving from one state to another over the edges given a specific action.

**Evaluation protocols** Besides the general graph generation metrics for directed graphs (degree, clustering, novelty, and uniqueness), we consider several dataset-specific metrics to evaluate node and edge generation quality.

1. *Edges*

    (a) MDP Validity (MV): Check if the sum of the outgoing edges of a node is one. It is a constraint of an MDP that the sum of probabilities is one. In the non-deterministic setup, the values are continuous, and therefore, we use an $\epsilon = 0.01$ gap from one. Note normalization of the edges could be done to fix this constraint if necessary. However, we evaluate the hard constraint to measure the model's ability to capture the edge feature distributions.

    (b) MDP Distribution Validity (MDV): As described before, the edge distribution is different in the non-deterministic setup. Therefore, we specifically test the model's ability to generate edges with approximately the same distribution. Consequently, we test whether each node's distribution we defined for the non-deterministic setup applies with an $\epsilon = 0.01$ gap.

2. *Nodes*

    (a) Valid Solution (VS): Measures if a generated grid is valid: has start and finish cells, and the route is not entirely blocked.

    (b) Blocks (B): Measures the absolute distance between the average number of blocks in the grids. In this work, the used dataset has a ground-truth average of four. For example, if a generated graph contains seven blocks, the distance will be 3, as the original distribution has only 4 blocks. The model is expected to match this target distribution.

    (c) Start and Finish (SF): Measures the absolute distance between the average number of end and start cells; there are always exactly two in the setup we used. For example, if a generated graph contains two starting cells and two finish cells, the distance will be 2, as the original distribution has only one for each, and the model is expected to match this target distribution.

    (d) Empty (E): Measures the absolute distance between the average number of regular cells. In this work, the dataset has a ground truth of 19 empty cells, and, similar to the previous two metrics, the generated graph should match this number.

### B.2.2 nuScenes Dataset

The nuScenes dataset (Caesar et al., 2020) serves as a crucial resource for trajectory prediction research in autonomous driving, featuring a vast collection of real-world sensor data recorded in the urban environments of Boston and Singapore. This dataset provides essential coordinates of vehicles, lanes, and other map entities from 1000 scenes, each lasting 20 seconds, and is published at 2Hz. Primarily, it is broadly used for trajectory prediction (Liu et al., 2021). Prior works used raster representations of the scenes combined with vision-based architectures to process the rasterized image. VectorNet (Gao et al., 2020) was the first to utilize a sparse graph representation of the scenes, where nodes represent agents and map elements, which are later processed via GNNs to predict the target. It has paved the way to its graph-based successors (Kim et al., 2021; Deo et al., 2022; Liu et al., 2023), now crowned as the current state-of-the-art that tops the leader-boards of this field.

Consequently, we follow their success and are convinced that such scenes can be naturally generated as graphs. For this purpose, we extract a portion of 746 samples of traffic scenarios from the *mini_train* split as defined in the nuScenes-devkit and use them for training and evaluation with 80%, 20% train, test split. Each sample is transformed into a graph with nodes of 3 types: 1. agents, with a feature vector representing an 8-second trajectory; 2. map elements, represented as x and y coordinates of their polygon; and 3. lanes, represented as discrete curves of length 8 meters each. We transform the graph into a radius graph of 30 meters and only preserve edges representing a relation whose target is an agent, e.g., lane-to-agent.

**Evaluation.** For evaluating the results for nuScenes graph generation, we use the standard protocol of evaluating the MMD over each node type, i.e., the $x$ and $y$ coordinates of the trajectories of vehicles (V), the coordinates of the lane curves (L) and other map objects (O). Additionally, we evaluate the Collision Rate (CR), which measures the rate of collisions between generated agents, and the Lane alignment (LA), which sums up the distances between each agent's trajectory to the closest generated lane. Such a metric reflects the tendency of road participants to follow lane center lines, which is a natural behavior of road participants. We refer to Tan et al. (2021) for more details about the metrics and evaluation protocols.

### B.2.3 Molecule Datasets

**QM9** - The Quantum Mechanics 9 database (Wu et al., 2018) contains around 130k small organic molecules with up to 9 heavy atoms and their physical properties in equilibrium, computed using density functional theory calculations. To evaluate our and other methods, we repeat the protocol presented in Vignac et al. (2023); Jo et al. (2022) and refer to them to learn more about the metrics and the evaluation protocols.

### B.2.4 General Datasets

We present the datasets below and refer to Vignac et al. (2023) and Jo et al. (2022) for more information about the evaluation process and protocols.

- **Ego-small** - 200 small ego graphs drawn from larger Citeseer network dataset (Sen et al., 2008).

- **Grid** - 100 standard 2D grid graphs. BRENDA database (Schomburg et al., 2004).

- **Stochastic-Block-Model (SBM)** - This dataset comprises 200 synthetic stochastic block model graphs. These graphs have communities ranging from two to five, with each community containing between 20 to 40 nodes. The probability of inter-community edges is 0.3, and intra-community edges is 0.05. Validity is determined based on the number of communities, the number of nodes in each community, and a statistical test as done in Martinkus et al. (2022).

- **Planar** - This dataset comprises 200 synthetic planar graphs, each containing 64 nodes. The criteria for a valid graph within this dataset necessitate a two-fold condition: 1) the graph is connected, ensuring that every node has a path to every other node, and 2) the graph must exhibit planarity, meaning you can draw it on a two-dimensional plane without any edge crossings.

### B.3 Limitations

Our approach exhibits superior performance compared to other methods on the MDP and nuScence datasets. However, there remains a quality disparity between the generated and original graphs. For example, in the MDP dataset, the generated graphs occasionally fail to adhere to the original constraints of start and end cells. We anticipate that an effective generative method would autonomously learn and adhere to such constraints. We believe that incorporating inductive bias regarding specific tasks can contribute to those tasks. Moreover, while score-based and diffusion methods typically demonstrate optimal performance across various downstream tasks in graph generation, scaling these methods to very large graphs containing millions of nodes and edges presents a challenge yet to be fully addressed.

### B.4 Implementation Details

### B.4.1 Architecture and Hyperparameters

We refer to Sec. 4 in the main text to describe the method architecture and implementation details. We present the hyperparameters per dataset in Tab. 4. We used Adam (Kingma & Ba, 2014) optimizer and the same learning rate of 0.01, weight decay of 0.0001, and EMA of 0.999 for all datasets. In addition, for sampling, we used the Euler predictor, Langevin corrector, signal-to-noise-ratio (SNR) of 0.05, scale epsilon of 0.7, and sampling with 1000 steps for all datasets. Finally, we use a single VP forward diffusion process (Song et al., 2021) stochastic differential equation for both the nodes and the edges. We refer to Jo et al.

Table 4: Hyperparameters for each dataset.

|  | Hyperparameter | MDP | MDP-non-det | nuScense | QM9 | Planar | SBM | Ego Small | Grid |
|---|---|---|---|---|---|---|---|---|---|
| Module | Attention layers | 5 | 5 | 3 | 3 | 4 | 4 | 5 | 4 |
|  | Edges channels | 4 | 2 | 1 | 2 | 2 | 2 | 2 | 2 |
|  | Initial channels | 1 | 1 | 1 | 1 | 1 | 1 | 1 | 1 |
|  | Hidden channels | 8 | 8 | 8 | 8 | 8 | 8 | 8 | 8 |
|  | Final channels | 4 | 4 | 4 | 4 | 4 | 4 | 4 | 4 |
|  | Attention Heads | 4 | 4 | 4 | 4 | 4 | 4 | 4 | 4 |
|  | Hidden dimension | 32 | 32 | 32 | 16 | 32 | 32 | 32 | 32 |
| Training | Batch size | 256 | 256 | 128 | 1024 | 64 | 26 | 128 | 7 |
|  | Epochs | 5000 | 5000 | 5000 | 400 | 5000 | 5000 | 5000 | 5000 |
| SDE | $\beta_{min}$ | 0.1 | 0.1 | 0.1 | 0.1 | 0.1 | 0.1 | 0.1 | 0.1 |
|  | $\beta_{max}$ | 3 | 3 | 2 | 1 | 1 | 1 | 1 | 1 |

(2022) for more details about each hyperparameter. We will publish the complete code, including the datasets, evaluation protocols, and experiment environment, upon acceptance.

### B.4.2 GDSS-E Baseline Implementation Details

GDSS (Jo et al., 2022) is a state-of-the-art method for graph generation. However, none of the other methods are adapted to generate multiple continuous edge features and directed graphs. Therefore, to create a solid baseline for directed, multi-edge attribute graph datasets, we create an extension of GDSS called GDSS-E. In this section, we describe GDSS and then how we adapt it to our context.

**GDSS diffusion framework.** In their paper, the authors adopt a general diffusion SDE modeling approach similar to that in Eq. 6. The main distinction from our framework is that they decompose the diffusion process into a system of SDEs (outlined in their paper Eq. 3). We highlight the advantages of our approach in Sec.4.1 and demonstrate its empirical improvements in the ablation study in Sec.5.1, where the joint SDE model shows enhanced capability over GDSS-E in learning the edge data distribution.

**GDSS architecture.** For a detailed explanation of their architectural choices, please refer to Section 3.2 of their paper. Briefly, due to the multiple SDE equation formulation, their model employs two separate Graph Neural Networks (GNNs): one for learning the score of the adjacency matrix and another for learning the node features. In contrast, our approach suggests using a single GNN to jointly model both graph components. Aside from this key difference, our architecture is constructed similarly to GDSS, with the addition of an edge-information propagation mechanism within the GNN's building block, as outlined in Sec. 4.2. Finally, we outline below additional differences and adjustments needed to adapt GDSS to the tasks and context of our paper.

**Directed graphs.** To adapt GDSS to a directed graph, we need to delete the symmetry inductive bias of the method. First, we deleted the symmetry inductive biases in the attention module of the backbone architecture. In addition, the generated noise and the whole diffusion process are set to be symmetric, meaning they generate symmetric noise patterns. Therefore, we change all aspects of the diffusion and generation process to be a normal Gaussian injection, similar to regular diffusion methods.

**Multiple edge features** We need to enable the model to generate multiple features technically. GDSS outputs an adjacency matrix $A \in \mathbb{R}^{N \times N}$ where $N$ is the number of nodes. We adjust the network parameter to return $E \in \mathbb{R}^{N \times N \times C}$ where the first feature is the adjacency information, and the rest of the channels are the attributes of the edges. We will publish this implementation code in the project code.

### B.4.3 SwinGNN-E Implementation Details

We adapt SwinGNN (Yan et al., 2023) to the edge-important benchmarks. We construct similar changes as we did for GDSS-E regarding directed graphs, and for multiple edge features, we modify the one-hot encoding configuration to work with the original data instead. We follow the same training protocols Yan et al. (2023) implemented for other similar datasets. We report the results in the main text.

### B.4.4 Permutation equivariance and invariance

Niu et al. (2020) show that if the neural backbone of a generative model is permutation equivariant, then the learned distribution by the model will be permutation invariant. This trait is essential for graph datasets since we ideally want an equal probability of sampling different permutations of the same graph. Following our model architecture, all our arithmetic actions are edge-wise, node-wise, or GNN-wise. Therefore, we preserve the permutation equivariance of the neural backbone model.

Yan et al. (2023) show that equivariance can be violated but restored with a specific invariant sampling technique. However, their study discovers one main drawback our and other permutation equivariant models do not suffer from. The drawback is that if graphs in the observed dataset have few permuted representations, it significantly damages the model generation quality. They showed on a synthetic dataset that if there are $\approx 0.01\%$ permuted representations, their model fails to learn the distribution. On the other hand, they show that equivariant models are indeed, as expected theoretically, robust for such cases.

## C   Additional Experiments and Analysis

### C.1   Time and Memory Comparison

In addition to the theoretical analysis discussed in the main text, we also conduct empirical evaluations regarding time and memory usage. We compare our model vs the GDSS-E baseline presented in the experiment section. By employing our innovative joint SDE rather than multiple SDE's like in GDSS, our models demonstrate superior performance in terms of both time efficiency and memory consumption compared to GDSS-E.

Table 5: Time and memory comparison. Time is the amount of time to train each model, both models trained on same device with the same seed and number of training epochs. Memory is the maximum memory consumption during the run.

| Method | nuScenes | | MDP-D | |
|---|---|---|---|---|
| | Time(Sec)↓ | Memory(MB)↓ | Time(Sec)↓ | Memory(MB)↓ |
| GDSS-E | $23,004$ | $2,122$ | $8,034$ | $2,485$ |
| Ours | $\mathbf{15,337}$ | $\mathbf{1,926}$ | $\mathbf{8,014}$ | $\mathbf{1,719}$ |

### C.2   Impact of increasing edge-feature size on adjacency matrix estimation.

Our method masks the adjacency matrix (Eq. 4), questioning the impact of increasing edge feature size on adjacency estimation. We found that feature distribution complexity, not size, may influence adjacency matrix topology. The Degree metric reliably compares adjacency matrices across different graph sizes, unlike the Cluster metric. In nuScenes, our method achieves a Degree score of 0.77, while in MDP, it's around five times smaller at 0.17. We hypothesize that this is due to nuScenes' more complex node feature distribution despite MDP having twice the number of edge features (2 vs. 4).

### C.3   Graph Benchmarks

Although we do not claim to have a superior distribution estimation for graphs where edge features are not necessary, and, in addition, our method is designed for directed graphs in contrast to all other methods, in

Table 6: General graphs datasets evaluation. '*' means out of computation resources.

| Method | Planar | | | SBM | | | Ego Small | | | Grid | | |
|---|---|---|---|---|---|---|---|---|---|---|---|---|
| | deg↓ | cl↓ | orb↓ | deg↓ | cl↓ | orb↓ | deg↓ | cl↓ | orb↓ | deg↓ | cl↓ | orb↓ |
| SPECTRE | 1.42 | 1.35 | 1.33 | 2.12 | 1.37 | 0.51 | 0.046 | 0.14 | 0.73 | * | * | * |
| GraphVAE | 0.87 | 1.13 | 0.83 | 1.41 | 0.97 | 0.52 | 0.13 | 0.23 | 0.052 | 1.48 | **0** | 0.87 |
| EDP-GNN | 0.985 | 1.29 | 0.97 | 1.1 | 1.43 | 0.88 | 0.062 | 0.097 | **0.009** | 0.45 | 0.32 | 0.51 |
| DiGress | 1.36 | 0.97 | 1.47 | 1.16 | 1.32 | 1.16 | 0.12 | 0.17 | 0.035 | 0.87 | 0.03 | 1.28 |
| GDSS | 0.945 | 0.96 | 0.66 | 0.74 | 1.57 | 0.25 | 0.025 | 0.087 | 0.015 | 0.37 | 0.01 | 0.42 |
| Our w.o. edge features | 0.032 | 0.71 | 0.34 | 0.47 | 1.1 | 0.05 | 0.02 | 0.043 | 0.052 | 0.07 | 0.012 | 0.45 |
| Our | **0.025** | **0.38** | **0.23** | **0.46** | **0.63** | **0.04** | **0.02** | **0.036** | 0.046 | **0.01** | 0.007 | **0.39** |

Table 7: Molecule QM9 dataset.

| Method | Val w/o↑ | Uni↑ | FCD↑ | NSPDK↓ |
|---|---|---|---|---|
| GDSS | 93.2% | 94.6% | 2.9 | 0.003 |
| DiGress | 95.5% | 94.1% | **0.578** | **0.0009** |
| Our | **96.7%** | **95.2%** | 3.6 | 0.006 |

the following two experiments, we compare our model to strong baselines on regular graph benchmarks. The generated edge features we use are arbitrary, and incorporating other engineered features on the edges, such as spectral features, could further improve our model. However, this is not our primary focus, and we leave such exploration for future research.

### C.3.1 General Graphs Benchmarks

To leverage the edge attributes ability of our model, we augment every graph with edge attributes per edge. Specifically, we compute the $n$-th power of the adjacency matrix, and then, for each edge $e_{ab}$ between nodes $a$ and $b$, we assign the corresponding value encoded in the power matrix. The edge features contain the number of paths between $a$ and $b$ with $n$ steps, where we set $n = 2$.

We compare our method with strong graph generation baselines: SPECTRE (Martinkus et al., 2022), GraphVAE (Simonovsky & Komodakis, 2018), EDP-GNN (Ho et al., 2020), DiGress Vignac et al. (2023), and GDSS (Jo et al., 2022). We follow the training and evaluation protocols detailed in Jo et al. (2022); Vignac et al. (2023). We present the results in Tab. 6. Our model achieves state-of-the-art performance in several cases. In particular, in complex and large graphs such as SBM, Planar, and Grid, our method is a strong competitor in terms of the *degree* metric compared with other methods. These results show that our method is capable of learning complex graph structures. Moreover, we report our results with standard deviation in App. 12 to show the robustness of our model. In addition, to make a fair comparison, we executed the benchmark with our model, excluding edge features. Our method achieves slightly lower results when edge features are not utilized. However, it still surpasses other methods in general.

### C.3.2 Molecule Graph Benchmark

**QM9: molecule generation.** We additionally consider the QM9 dataset (Wu et al., 2018) that contains edge types and node features of atoms of molecules. We refer the reader to App. B.2.3 to learn about the dataset, evaluation protocol, and metrics. We report in Tab. 7 the results of our evaluation in comparison to GDSS and DiGress. Importantly, we emphasize that DiGress is explicitly designed to handle the generation of edge types as it leverages transition kernels. Nevertheless, our method shows strong results, achieving the best scores on the validity w/o (Val w/o) and Uniqueness (Uni) metrics.

## C.4   General Graphs Ablation Study Cont.

We extend the ablation study over the general graph benchmarks presented in Sec.5.5 and report the results in Tab.8. The results support the claimed contributions of our model components, as presented in the main text.

Table 8: Ablation study of our four variants of our model on four different datasets.

| Method | Ego Small | | | Grid | | |
|---|---|---|---|---|---|---|
| | deg↓ | cl↓ | orb↓ | deg↓ | cl↓ | orb↓ |
| GDSS-E | 0.141 | 0.352 | 0.171 | 0.37 | 0.01 | 0.42 |
| Joint-SDE-Model | 0.063 | 0.167 | 0.062 | 1.8 | **0** | 1.44 |
| GNM-Based-Model | 0.021 | 0.038 | 0.048 | 0.49 | 0.006 | 0.51 |
| Ours | **0.04** | **0.02** | **0.036** | **0.046** | **0.01** | 0.007 |

## C.5   Standard Deviation in Experiments

We present the results of the quantitative evaluations with standard deviation to emphasize our method's robustness. We show the MDP deterministic setting in Tab. 9. We show the MDP non-deterministic setting in Tab. 10. In Tab. 11, we show nuScenes. In Tab. 12, we show the general graphs experiment with standard deviation.

Table 9: Deterministic MDPs with standard deviation.

| Method | deg ↓ | cl ↓ | un↑ | no↑ | MV ↑ | VS ↑ | B ↓ | SF ↓ | E ↓ |
|---|---|---|---|---|---|---|---|---|---|
| GDSS-E | $0.73 \pm 0.025$ | $0.06 \pm 0.002$ | $97 \pm 1$ | $100 \pm 0$ | $34\% \pm 1\%$ | $9\% \pm 2\%$ | $0.96 \pm 0.16$ | $1.28 \pm 0.05$ | $2.23 \pm 0.16$ |
| Our | $\mathbf{0.17 \pm 0.02}$ | $\mathbf{0.006 \pm 0.001}$ | $100 \pm 0$ | $100 \pm 0$ | $\mathbf{68\% \pm 1\%}$ | $\mathbf{34\% \pm 4\%}$ | $\mathbf{0.1 \pm 0.04}$ | $\mathbf{0.58 \pm 0.13}$ | $\mathbf{0.48 \pm 0.18}$ |

Table 10: Non-deterministic MDPs with standard deviation.

| Method | deg ↓ | cl ↓ | un ↑ | o↑ | MV ↑ | MDV ↑ | VS ↑ | B ↓ | SF ↓ | E ↓ |
|---|---|---|---|---|---|---|---|---|---|---|
| GDSS-E | $0.40 \pm 0.02$ | $0.02 \pm 0.01$ | $99 \pm 1$ | $100 \pm 0$ | $6\% \pm 1\%$ | $1\% \pm 0.5\%$ | $26\% \pm 2\%$ | $0.39 \pm 0.03$ | $\mathbf{0.83 \pm 0.1}$ | $\mathbf{0.4 \pm 0.1}$ |
| Our | $\mathbf{0.31 \pm 0.01}$ | $\mathbf{0.01 \pm 0.001}$ | $100 \pm 0$ | $100 \pm 0$ | $\mathbf{38\% \pm 2\%}$ | $\mathbf{6\% \pm 0.5\%}$ | $\mathbf{33\% \pm 1\%}$ | $\mathbf{0.02 \pm 0.02}$ | $0.88 \pm 0.1$ | $0.8 \pm 0.01$ |

Table 11: Quantitative metrics on nuScenes with standard deviation.

| Method | deg↓ | cl↓ | un ↑ | no↑ | V↓ | O↓ | L↓ | CR↓ | LA↓ |
|--------|------|-----|------|-----|-----|-----|-----|-----|-----|
| GDSS-E | $1.05 \pm 0.1$ | $0.03 \pm 0.003$ | $15 \pm 5$ | $34 \pm 2$ | $3.9 \pm 0.19$ | $\mathbf{0.66 \pm 0.13}$ | $0.96 \pm 0.14$ | $0.5\%$ | $208 \pm 21$ |
| Our | $\mathbf{0.77 \pm 0.08}$ | $\mathbf{6e^{-7} \pm 4e^{-7}}$ | $51 \pm 3$ | $51 \pm 3$ | $\mathbf{0.36 \pm 0.01}$ | $0.8 \pm 0.1$ | $\mathbf{0.08 \pm 0.03}$ | $\mathbf{0.3}\%$ | $\mathbf{194 \pm 15}$ |

Table 12: General graph datasets evaluation with standard deviations of our method

| Dataset | degree ↓ | cluster ↓ | orbit ↓ |
|---------|----------|-----------|---------|
| Planar | $0.025 \pm 0.009$ | $0.38 \pm 0.06$ | $0.23 \pm 0.005$ |
| SBM | $0.46 \pm 0.09$ | $0.63 \pm 0.04$ | $0.05 \pm 0.0001$ |
| Ego Small | $0.02 \pm 0.011$ | $0.036 \pm 0.0087$ | $0.046 \pm 0.0073$ |
| Grid | $0.01 \pm 0.003$ | $0.007 \pm 0.001$ | $0.39 \pm 0.07$ |

## C.6   Graphs Visualizations Generated by Our Model

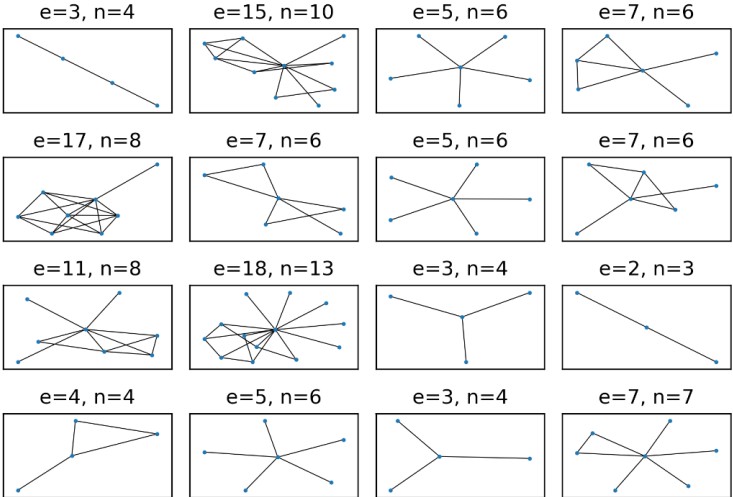

Figure 8: General graphs - Ego Small

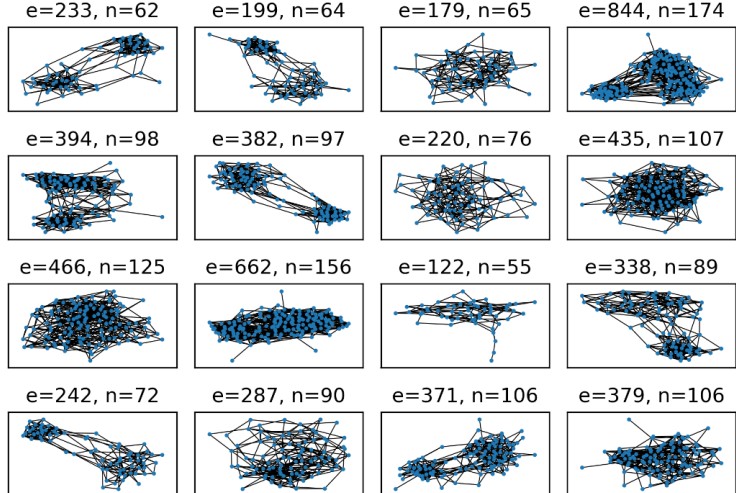

Figure 9: General graphs - SBM

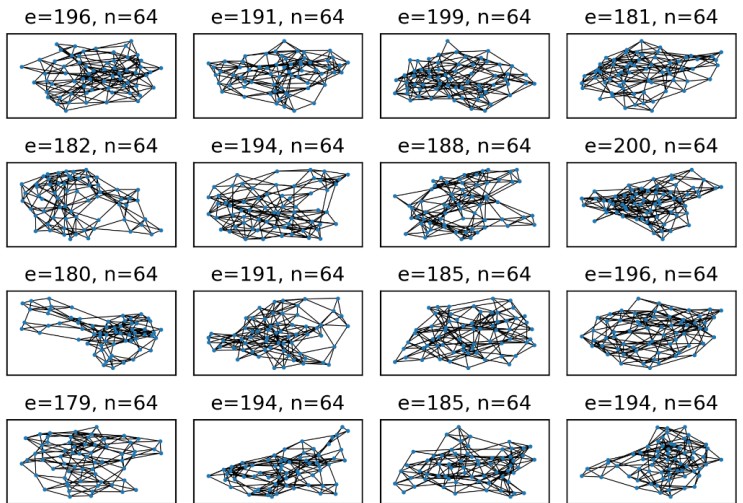

Figure 10: General graphs - Planar

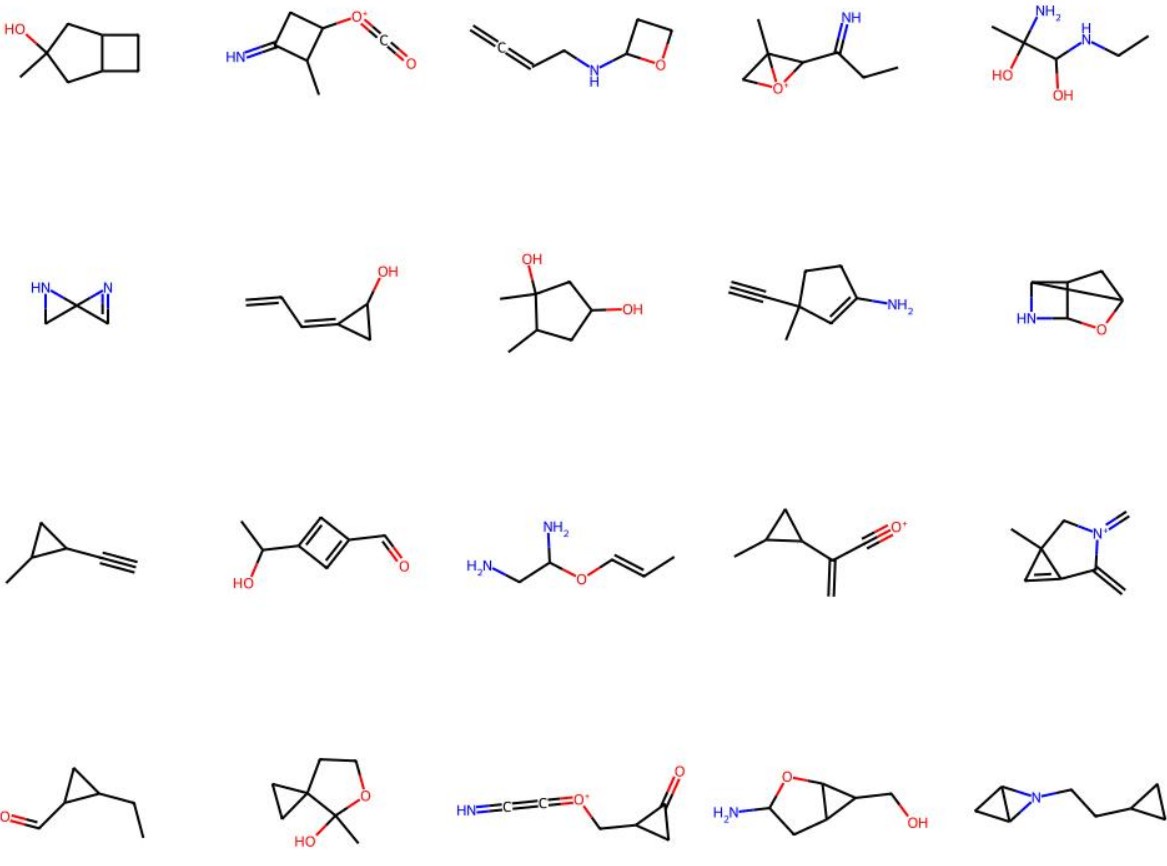

Figure 11: Molecule Graphs - QM9

