# OpenReview forum: "Reviving Life on the Edge: Joint Score-Based Graph Generation of Rich Edge Attributes"
_TMLR — Accepted by TMLR_

### Review · Reviewer_ZnJ5 · 2024-09-20

**Summary Of Contributions:**

This paper introduces a novel method for graph generation. It addresses the specific problem of generating graphs with continuous edge features, something that was until then not sufficiently addressed in the literature. The approach relies on a diffusion-based generation process, with specific modules specializing in handling edge features, and allowing inter-dependency between edge features and node features.

The authors then provide extensive experiments showing the performance of their model on different tasks, such as generating Markov decision processes, molecular graphs or urban driving scenarios, as well as ablation studies to show the specific strengths and weaknesses of their two components: a score based on both edges and node features, and a specific attention-based architecture for learning inter-dependency among edge and node features.

**Audience:**

Yes

**Claims And Evidence:**

No

**Requested Changes:**

In reference to the "Major" weaknesses in the previous section:

* Polishing the paper overall, and specifically section 5 (making technical points very clear) is necessary for the paper to be non-ambiguous on several points. All critical information should be in the main paper rather than the appendix.
* The metrics for evaluating the generated graphs are relevant and good, but not sufficient to show the good performance of the approach as a generative model. The analysis of the results lacks a study of diversity and novelty of the generated graphs. Adding such metrics is necessary, I think, for for guaranteeing that the claims of the paper are fulfilled.
* Addressing the "minor" weaknesses above would strengthen the work.

**Strengths And Weaknesses:**

This paper introduces a novel method for graph generation. It addresses the specific problem of generating graphs with continuous edge features, something that was until then not sufficiently addressed in the literature. The approach relies on a diffusion-based generation process, with specific modules specializing in handling edge features, and allowing inter-dependency between edge features and node features. The authors then provide extensive experiments showing the abilities of their model on different tasks, such as generating Markov decision processes, molecular graphs or urban driving scenarios, as well as ablation studies to show the specific strengths and weaknesses of their two components: a score based on both edges and node features, and a specific attention-based architecture for learning inter-dependency among edge and node features.


# Strengths:
* The problem addressed by the paper, namely generation of graphs with rich, non-discrete edge features is a highly interesting one. It is true that edge features carry crucial information in a wide array of cases.
* The method presented is relevant to the problem at hand. The architecture is coherent with the claims of the paper.
* Section 4 presents the approach in a mostly clear way (although some clarifications could be made, see below).
* Most of the content necessary for understanding is present, and some finer details are provided to answer basic interrogation that a reader could have.
* The approach is tested on a large number of experiments, especially very relevant ablation studies, and seem to yield promising results (which need to be completed, see below).
* The approach is (as far as I know) novel and original.

# Weaknesses:
## Major:

* Some crucial metrics are lacking in the results. The metrics presented only measure the generated graphs' similarity to the training set. Yet we could expect measures of:
  * Novelty: showing that the model generates graphs that are not present in the training set
  * Diversity: showing that the model generates a large number of diverse graphs and not only graphs that are very similar to one another (i.e. some kind of mode collapse). The standard deviations (which are only provided in the appendix) do not show diversity.
Without those, the model presented could consistently generate only a handful of preexisting graphs without this appearing in the current metrics. Of course, some qualitative illustration could complement this (such as some examples of generated graphs in appendix, like it was done for EGO, SBM and QM9, but for the other cases).

* While most of the information required is "there", the writing and presentation are sometimes quite confusing. This hinders a quick and efficient reading of the paper. Some entire sections look like a draft. Examples include:
  * Some concepts/acronyms are used before being explicited (e.g. SDE, MDP, MMD)
  * Some results and important points are in the Appendix, or missing, while they should be in the main text (e.g. Appendix B.1, metrics variances, clean experimental tasks description);
  * Some notions are not described formally enough, leading to ambiguity for things that should be clear:
    * homogeneity in 5.1-evaluation: "edges whose both features belong to only one of the cluster" is ambiguous;
    * equation 8 uses some implicit broadcasting on the edge-related term, which leads to ambiguity. Maybe the dimensions of both weight tensors should be given to make everything clear immediately
    * What is f in practice in eq 5? Thus what is in practice the limit distribution (at time T) in the diffusion process (from which you sample for generating)?
  * Section 5.2 (and the associated appendix B.2.1) is not well written. It is very laborious to read, especially since lot of critical information (e.g. definition of the features, of the data) is in the appendix which is badly written (almost looks like a draft sometimes). There are also inconsistencies in the terminology (e.g. obstacles are called blocks in the appendix). Please clarify especially clearly and concisely what the node features, edge features and constraints are, but I think that a strong polishing of section 5.2 and App.B.2.1 is needed.
* The relevance and robustness of parameter epsilon for equation 4 is never addressed. Nonetheless, since equation 4 is not differentiable and not part of the training process, it seems that the model has no reason to "learn" that there will be an edge iff this criterion is met. It thus seems really arbitrary, and could lead to absurd results in many cases (for instance if the learned features are slightly shifted, you could easily go from a complete graph to a graph with no edge despite very little change in the learned model). An analysis of this criterion's role could be suitable, and mechanism for avoiding instability.

## Minor:
Below are the minor concerns.
* GDSS is evoked and used as the main comparison baseline in this paper. Some more formal details as to how it works and its differences with this approach would be highly relevant (even if in the appendix).
* Most figures rely only on color to differentiate certain crucial elements. This should be avoided as much as possible. Using shapes or symbol will significantly improve accessibility.
* Both halves of figure 2 should be more clearly separated, and the fact that the right is a "detailed view" of the element in the left could be made slightly clearer (textual indication in the figure for instance).
* All equations are written as "Eq. equation"
* 5.1: what do you mean "sampled randomly from only one of the clusters"? Please detail this (cf. the "homogeneity" comment above)
* from the described process, it seems that the number of nodes of the generated graph is fixed in advance, right? Even though the paper focuses on generating edge features, this is a strong limitation which should be addressed, I think.
* some redundancy could be avoided between the introduction and related works, as many works are cited in both.

---

> ### Author Response · Authors · 2024-11-15
> **Response 1**
>
> > Some crucial metrics are lacking in the results. The metrics presented only measure the generated graphs' similarity to the training set. Yet we could expect measures of:  Novelty: showing that the model generates graphs that are not present in the training set  Diversity: showing that the model generates a large number of diverse graphs and not only graphs that are very similar to one another (i.e. some kind of mode collapse)...
>
> Please see the comment on the metrics below, where we cited the SPECTRE approach.
>
> > Some concepts/acronyms are used before being explicited (e.g. SDE, MDP, MMD)
>
> Thank you. Our revised version fixed these issues.
>
> > Some results and important points are in the Appendix, or missing, while they should be in the main text (e.g. Appendix B.1, metrics variances, clean experimental tasks description)
>
> We have made our best to incorporate all relevant information into the main paper as suggested in this and other comments and marked in red in the revised version, while adhering to the 12-page limit. We hope that the new revision will significantly improve the clarity and we would be happy to address any further comments or make additional changes if needed. For the reviewers' convenience, we have uploaded a revised version with changes marked in red. Briefly, we clarified experimental setups and metrics definition and moved essential information from the appendix to the main text.
>
> > homogeneity in 5.1-evaluation: "edges whose both features belong to only one of the cluster" is ambiguous
>
> Thank you for bringing this to our attention. In our revision, we have clarified this point by specifying that all edges in the graph should belong exclusively to one of the clusters, consistent with the structure of the original data. Changes can be seen in red marker.
>
> > equation 8 uses some implicit broadcasting on the edge-related term, which leads to ambiguity. Maybe the dimensions of both weight tensors should be given to make everything clear immediately
>
> We have enhanced Eq.~8 by introducing a $\textrm{rep}(\cdot)$ operator, which replicates its input $v$ times along the third dimension. This modification eliminates the need for complex broadcasting operations, simplifying the notation.
>
> > What is f in practice in eq 5? Thus what is in practice the limit distribution (at time T) in the diffusion process (from which you sample for generating)?
>
> We employed the variance preserving stochastic differential equation, where $f(G,t) = -\frac{1}{2}\beta(t) G$. The schedule for $\beta$ is linear, interpolating between high and low beta values, leading to an approximate limit distribution of $\mathcal{N}(0, I)$ at time $T$.
>
> > Section 5.2 (and the associated appendix B.2.1) is not well written. It is very laborious to read, especially since lot of critical information (e.g. definition of the features, of the data) is in the appendix which is badly written (almost looks like a draft sometimes). There are also inconsistencies in the terminology (e.g. obstacles are called blocks in the appendix). Please clarify especially clearly and concisely what the node features, edge features and constraints are, but I think that a strong polishing of section 5.2 and App.B.2.1 is needed.
>
> Thank you for highlighting these points to help improve this section. We have revised the main text to clarify ambiguous details and expanded the appendix to enhance clarity. All changes are marked in red in the uploaded file for your convenience. In summary, we have standardized the terminology by consistently referring to block cells as 'blocks' rather than 'obstacles,' added clarifications on metric calculations, and provided further explanation on the structure of nodes, edges and constraints in both the main text and the appendix.
>
> > The relevance and robustness of parameter epsilon for equation 4 is never addressed. Nonetheless, since equation 4...
>
> Thank you for raising this point. The threshold parameter $\epsilon$ in Equation 4 is indeed static, but we find it suitable for our chosen datasets. For MDPs, low-probability transitions are naturally disregarded, while in traffic generation, small distances represent invalid links. Thus, $\epsilon$ serves to remove connections lacking significance.
>
> Additionally, we note that $\epsilon$ could generalize across datasets with appropriate data normalization, making it adaptable beyond our specific tasks. For future work, we could explore an adaptive $\epsilon$ to further enhance stability. We have added a brief explanation to the main text, to further give more intuition for the choice of this term.

---

> ### Author Response · Authors · 2024-11-15
> **Response 2**
>
> > GDSS is evoked and used as the main comparison baseline in this paper. Some more formal details as to how it works and its differences with this approach would be highly relevant (even if in the appendix)
>
> Following your suggestion, we have expanded the subsection "GDSS-E Baseline Implementation Details" in the appendix to include information about GDSS and added description of how it is different from our model.
>
> > Most figures rely only on color to differentiate certain crucial elements. This should be avoided as much as possible. Using shapes or symbol will significantly improve accessibility.
>
> Thanks for the comment. We have incorporated symbols as suggested in the new revision.
>
> > Both halves of figure 2 should be more clearly separated, and the fact that the right is a "detailed view" of the element in the left could be made slightly clearer (textual indication in the figure for instance).
>
> We improved Fig. 2 as suggested by the reviewer.
>
> > All equations are written as ``Eq. equation''
>
> Thank you. We fixed this issue in our revision.
>
> > 5.1: what do you mean "sampled randomly from only one of the clusters"? Please detail this (cf. the "homogeneity" comment above)
>
> We added a sentence, specifying that graph edges are homogeneous, i.e., they are sampled from the same cluster.
>
> > From the described process, it seems that the number of nodes of the generated graph is fixed in advance, right? Even though the paper focuses on generating edge features, this is a strong limitation which should be addressed, I think...
>
> You're correct that our current method requires a fixed number of nodes to be specified in advance. Yet, the graph size is a property which can be easily sampled from the distribution of the dataset and thus doesn't necessarily has to be decided by the generation process, but rather be a conditioning parameter that is sampled in advance. And yet, this property is shared by many existing graph generation approaches such as GDSS itself and swinGNN, which also typically require the number of nodes to be predefined. Thus, this is not unique to our approach, and extending our method to support variable node generation is indeed an interesting direction and something we consider a priority for future work. Our current focus is on generating edge features to advance representational accuracy, but we will include this as a future research direction.
>
>
> > some redundancy could be avoided between the introduction and related works, as many works are cited in both.
>
> After further considerations regarding the final content, we decided to move some content from the introduction to the related works section.
>
> > Polishing the paper overall, and specifically section 5 (making technical points very clear) is necessary for the paper to be non-ambiguous on several points. All critical information should be in the main paper rather than the appendix.
>
> In response to the reviewers' comments and suggestions, we have refined the paper, particularly Section 5. Due to space constraints, we focused on keeping the most essential information in the main text. An anonymous version with changes marked in red has been provided for your convenience.
>
> > The metrics for evaluating the generated graphs are relevant and good, but not sufficient to show the good performance of the approach as a generative model. The analysis of the results lacks a study of diversity and novelty of the generated graphs. Adding such metrics is necessary, I think, for for guaranteeing that the claims of the paper are fulfilled.
>
> Thank you for bringing this to our attention. We have followed your suggestion and evaluated the models using additional metrics. Following [1], we adapted the novelty and uniqueness metrics to fit our benchmark and have included the updated results in the revised paper, with changes highlighted in red in the table. In summary, all models perform comparably well on the MDP dataset; however, on the more complex Nuscenes dataset, our method outperforms the others.
>
> [1] SPECTRE: Spectral Conditioning Helps to Overcome the Expressivity Limits of One-shot Graph Generators.
>
> > Addressing the "minor" weaknesses above would strengthen the work.
>
> We addressed all minor issues above, and we would be happy to address any remaining concerns.

---

### Review · Reviewer_v9yH · 2024-10-15

**Summary Of Contributions:**

This paper proposes a generative model of graphs with a focus on edge attributes. The proposed model is based on a score-based model, and has two main features. One is that it jointly models the diffusion process of the nodes and adjacency matrix (while GDSS models separate diffusion processes for them).  The other feature is a graph neural module (GNM), which computes a node-wise latent representation by combining both node and edge attributes.

The authors demonstrate the effectiveness of the proposed method by experiments.

The first experiment is an ablation study using a synthetic dataset. The result demonstrates that both features are essential for reliable graph generation with edge features, suggesting that the joint SDE process is essential for inter-edge interactions, while the GNM is essential for intra-edge interactions.

The second experiment imposes a maze MDP generative task. The authors make use of several statistics of a maze environment to see whether those of the generated maze match with those of the ground truth one. The result shows that the proposed method could generate more realistic mazes than the others in terms of the statistics.

The third experiment uses a traffic scene generation task, where the relationship between cars, tracks, traffic lights, and others is encoded into a graph. In specific, a node corresponds to an entity and the edge feature represents the Euclidean distance between the two entities. The result also shows the effectiveness of the proposed method in many metrics.

**Audience:**

Yes

**Claims And Evidence:**

Yes

**Requested Changes:**

# Typos
- Equations are referred by "Eq. equation 1", which is likely to be due to a misuse of a LaTeX command \eqref. Please revise it.

# Clarity

I would request the authors to improve the clarity at least by addressing the concerns raised in the weakness, but desirably by reviewing the whole manuscript again and fixing unclear statements.

**Strengths And Weaknesses:**

# Strengths

The experiments are well designed with clear statements of their objectives. Except for the details of the metrics employed, experimental procedures, results, and discussions are mostly reasonable.



# Weaknesses

A major weakness is that there are incorrect or unclear statements, and the manuscript needs to be revised by improving the clarity. In the following, I will provide several examples, but I would like to request the authors to review the manuscript again and improve the clarity.

In Equation 8, $\bar{A}_t$ is a $n\times n$ matrix, $E_t$ is a $n\times n\times v$ tensor, and the element-wise product between them is mathematically illegal.

Metrics used in Section 5.2 are not easy to understand, possibly due to grammatical errors. For example, given the following sentence,
> "Start and finish (SF) calculates the distance between the average number of start and finish cells, which are always two in our grids."
it is not clear i) which distance the authors use and ii) what is the distance between the average number of start and finish cells. If the sentence were "the distance between the average numbers of start and finish cells" or "the distance between the average number of start and finish cells of generated mazes and that of ground truth ones", it would be clear except for which distance were used.

Metrics in Section 5.3 are also not easy to understand. Most of the explanations are excluded from the main text, and are moved to the appendix, although these are essential to understand the experimental results. In addition, the explanations are not clear. For example, MMD is used without its explanation (does it stand for the maximum mean discrepancy?).

---

> ### Author Response · Authors · 2024-11-15
> **Response 1**
>
> > A major weakness is that there are incorrect or unclear statements, and the manuscript needs to be revised by improving the clarity...
>
> Thank you for highlighting key points and giving us the opportunity to enhance our paper quality. Below, we address each of your specific comments. In general, following the reviewers' feedback, we have made substantial improvements to the paper. For your convenience, we have uploaded an anonymous revision with changes marked in red. In summary, we have addressed all editorial comments, clarified experimental setups and metrics, clarified differentiations between methods, added new metrics.
>
> > In Equation 8, $\bar{A}_t$ is a matrix, $E_t$ is a $n\times n \times v$ tensor, and the element-wise product between them is mathematically illegal.
>
> Thank you. We have enhanced Eq.~8 by introducing a $\textrm{rep}(\cdot)$ operator, which replicates its input $v$ times along the third dimension. This modification eliminates the need for complex broadcasting operations, simplifying the notation.
>
> > Metrics used in Section 5.2 are not easy to understand, possibly due to grammatical errors. For example, given the following sentence…
>
> We have refined Section 5.2, making the text clearer and more concise. We've clarified the start-finish definition and added examples for better understanding. Additionally, we expanded the dataset discussion and introduced new metrics to benchmark uniqueness and novelty as suggested by other reviewers.
>
> > Metrics in Section 5.3 are also not easy to understand. Most of the explanations are excluded from the main text, and are moved to the appendix, although these are essential to understand the experimental results.
>
> We have clarified the previously unclear MMD text, specifying that it stands for maximum mean discrepancy. Additionally, we moved the metric explanations from the appendix to the main text. Additionally, we have added novelty and uniqueness metrics.
>
> > Equations are referred by "Eq. equation 1", which is likely to be due to a misuse of a LaTeX command \eqref. Please revise it.
>
> Thank you, we have fixed this issue.

---

> ### Comment · Reviewer_v9yH · 2024-11-24
> **Re: Response 1**
>
> Thank you very much for updating the manuscript. I confirmed my concerns had been addressed (but still there are many typos; for example, Eqs. equation (1) and equation (2) in the second line from the bottom of page 5). I am ok to accept this paper, but I would strongly encourage the authors to carefully review the manuscript again and fix typos and grammatical errors.

---

> > ### Author Response · Authors · 2024-11-24
> > **Thank you**
> >
> > Thank you very much for your review, comments, and recommendation for the acceptance of the paper.
> >
> > We will conduct another detailed review of the manuscript to correct the equation formatting and resolve any remaining grammatical or typographical errors.

---

### Review · Reviewer_TbWc · 2024-10-23

**Summary Of Contributions:**

This paper addresses the challenge of modeling graph generation as a diffusive process. The primary claim is that previous approaches fall short in jointly modeling all graph components, particularly integrating both node and edge attributes within a single diffusive step.
 The proposed method is evaluated on a traffic scene generation task, as well as synthetic tasks involving edge attribute generation on complete graphs and grid mazes.

**Audience:**

Yes

**Claims And Evidence:**

No

**Requested Changes:**

Please see the weakness stated above.

**Strengths And Weaknesses:**

Strengths

The paper addresses an important and challenging problem in the field of graph generation, which is a key contribution to the domain.
The approach of modeling the complete graph generation process end-to-end is particularly intriguing and shows promise for advancing current methods.

Weaknesses

Lack of clear motivation for the proposed method:
While the paper argues that “edge attributes are essential in various domains where information cannot be encoded otherwise,” it doesn’t clearly illustrate which features cannot be represented as node features. This weakens the argument that previous models lack expressiveness. For instance, in the driving scene example, relative velocity, acceleration, and distance could be derived from node features. Including simple toy examples to support this claim would help strengthen the paper’s motivation.

Presentation of the method:
The technical contribution, namely the joint SDE model, is introduced without sufficient discussion of the key considerations and challenges involved in designing this particular model. For example, the model appears to parameterize all possible connections  E_{ijk} , despite the fact that most real-world graphs are sparse. Does this create limitations for the model? Is this the reason previous works haven’t used this approach?

Moreover, the advantage of predicting the adjacency matrix  $A$  via $E_{ijk}$  isn’t made clear. How does this compare to using a simpler approach like  $A = \sigma(X X^t)$ , where  $X$  represents node features?

Similarly, the GNN architecture is not well-justified. Are there any specific challenges posed by the diffusion problem that necessitate the architectural choices made in this model?

Insufficient discussion of graph topology generation:
In the first toy experiment, complete graphs are used, but it’s not clear how this experiment demonstrates the unique contributions of the proposed approach. What is the advantage over running a standard diffusion model with an MLP, where each element is a concatenation of all edge information? More discussion is needed to clarify the differences and justify the method’s novelty.

---

> ### Author Response · Authors · 2024-11-15
> **Response 1**
>
> > Lack of clear motivation for the proposed method: While the paper argues that “edge attributes are essential in various domains where information cannot be encoded otherwise,” it doesn’t clearly illustrate which features cannot be represented as node features. This weakens the argument that previous models lack expressiveness. For instance, in the driving scene example, relative velocity, acceleration, and distance could be derived from node features. Including simple toy examples to support this claim would help strengthen the paper’s motivation.
>
>   In dense traffic, drivers rely on subtle, situational cues to negotiate maneuvers. While some relative features mentioned in the paper can clearly be calculated from node features, the “looking at” edge feature, a flag indicating if one vehicle is actively observing another, captures this real-time, mutual awareness. For example, during a lane change, one driver may watch another for cues on their intent, like yielding or maintaining position. This relational feature can’t be inferred from node attributes alone, as it represents a temporary, dynamic focus that shifts based on interactions, underscoring the need for edge features to accurately model these driving behaviors. Additionally, for the second use cases of representing Markov Decision Processes as graphs, non-inclusion of edge features limits expressiveness for capturing nuanced relationships. In an MDP, edge features like transition probabilities and rewards are crucial, as they encode the dynamics of moving from one state (node) to another based on actions. These values are often unique to each transition, representing not just a link but a meaningful interaction that impacts decision-making. Without edge features, we are forced to encode transition information at the node level, leading to redundancy and inflating graph complexity. By leveraging edge features, we create a compact, accurate representation of the MDP, where each edge directly encodes the transition’s probability and reward, preserving the true structure of the decision process without compromising graph simplicity.
>
> We thank you for the opportunity to address this matter and in the revised version, we have expanded the discussion on the motivation for the graph representation in the main text introduction section. We have uploaded revised version with changes marked in red.
>
> > ... For example, the model appears to parameterize all possible connections $E_{ijk}$ , despite the fact that most real-world graphs are sparse. Does this create limitations for the model?
>
> Thank you for your question. The diffusion generative framework can be applied to any data distribution, including all types of graph data distributions. Specifically, our edges 'E' modeling is learnable and can learn from the observed data. Consequently, our method can effectively capture the data distribution of any graph type, whether sparse or dense and there is no limitation regarding the input data. Finally, while targeting sparse graphs can improve computational complexity, it necessitates incorporating data- or task-specific biases into the modeling process. Instead, we have chosen a general approach that avoids making any assumptions about the data or task, ensuring broader applicability and flexibility.
>
> > ... Is this the reason previous works haven’t used this approach?
>
> Previous methods did not aim to generate graphs with rich edge features; instead, they focused on a simpler setup where edges contained either no information or only a single discrete label. Thus, previous work fundamentally addressed a different problem with a distinct objective, which likely resulted in the development of entirely different approaches in practice.
>
> > Moreover, the advantage of predicting the adjacency matrix A via Eijk isn’t made clear. How does this compare to using a simpler approach like $A=\sigma(XXt)$ , where X represents node features?
>
> In both our setup and that of previous work, a node in $X$ is a multidimensional vector that can include various features, such as age or gender in a social graph. Calculating connectivity based solely on the product of two node features is not a reliable indicator of a connection. In particular, the computation $X X^T$ effectively ``creates'' edges $e_{ij}$ in all instances where nodes $i$ and $j$ have non-zero features, irrespective of the actual relationships between those nodes. Thus, it is essential to extract adjacency information from $E$, which encodes the details of the graph's edges. If an edge contains information, a connection will be established; otherwise, no connection will be determined.

---

> ### Author Response · Authors · 2024-11-15
> **Response 2**
>
> > Similarly, the GNN architecture is not well-justified. Are there any specific challenges posed by the diffusion problem that necessitate the architectural choices made in this model?
>
> In our context, the diffusion problem is formulated as a stochastic differential equation (SDE) where we aim to learn the reverse diffusion process. This reverse process depends on a single unknown function, the score function, as detailed in the background section of our paper. The score function can theoretically be modeled and learned by any neural network and is referred to as the score model. For graph data specifically, both our work and prior methods use a permutation-equivariant score-based model, which has been shown to capture permutation-invariant distributions--a desirable property for learning graph data distributions. Thus, the diffusion framework itself does not constrain architectural choices, though the graph data type necessitates using GNNs. We are happy to provide further details or additional information if needed.
>
> > Insufficient discussion of graph topology generation: In the first toy experiment, complete graphs are used, but it’s not clear how this experiment demonstrates the unique contributions of the proposed approach.
>
> Thank you for pointing this out and allowing us to clarify. The first experiment analyzes the models' performance in accurately generating edge features, as other components of the graph (connectivity and node features) are straightforward for all models to generate. Although we use complete graphs in this toy example, the graphs include specific edge features. Consequently, beyond generating correct connectivity, the compared methods must also effectively capture the distribution of these edge features. Our experiment demonstrates that other approaches struggle to accurately capture the feature distribution of the graph edges. To further clarify, we have added this explanation to the revised version.
>
> > What is the advantage over running a standard diffusion model with an MLP, where each element is a concatenation of all edge information? More discussion is needed to clarify the differences and justify the method’s novelty.
>
> A standard MLP or U-Net does not maintain the permutation-invariance property required for data generation. Permutation equivariance is crucial when modeling graphs because the labeling of nodes in a graph is arbitrary, and the underlying structure and relationships must remain invariant under any reordering of nodes. This property ensures that the model's predictions depend only on the graph's topology and features, not on the specific order in which nodes or edges are presented, leading to more robust and generalizable representations. In Appendix Section B.4.4, we provide further details on how our model preserves this property.

---

### Author Response · Authors · 2024-11-15
**General Response**

Dear reviewers,

Thank you for taking the time to provide detailed feedback on our work. We appreciate your recognition of the importance of the problem we address, the novelty of our approach, and the breadth of our empirical experiments. We are also deeply grateful for your constructive comments, which have greatly helped us improve the quality of the paper.

We have addressed each reviewer’s comments and concerns in their respective threads. Additionally, for your convenience, we have uploaded a revised version, with all changes marked in red.

Thank you once again for your valuable input.

---

### Decision · Action_Editor_JAq9 · 2024-12-04

**Recommendation:** Accept with minor revision

**Comment:**

The paper was reviewed by three expert reviewers. All reviewers expressed significant concerns about the clarity of presentation. The authors revised the manuscript and improved the clarity, and two reviewers recommended weak acceptance of the paper, while one reviewer recommended weak rejection.

It is also my view that the paper could be further improved in terms of clarity. A "minor revision" is thus recommended for the paper, and I request that the final version of the manuscript considers the reviewers' comments, as well as the following:

- Please provide intuition for the different components of the proposed model.
- Revise subsection 4.2.
    - In Equation (8), what is $tanh(B[rep(\tilde{A}_t) \odot E_t W_E ])$ expected to learn? Please provide more details.
    - What is the output of the GNM module? Is it an $n \times d$ matrix that stores the new representations of the nodes?
    - What is the output of the ATTN module? Is it an $n \times n$ matrix that stores the attention scores for the different pairs of nodes?
    - In Equations (10) and (11), the outputs of $J$ different GNM and ATTN modules are concatenated and fed to an MLP. Are those GNM and ATTN modules just $J$ randomly initialized instances of those modules?
    - Equations (10) - (15) need to be further discussed.
- Fix issues with parenthetical and narrative citations (e.g., first paragraph of subsection 4.2).
- Fix typos and grammatical errors (e.g., in page 5, fix the following: "in Eqs. equation 1 and equation 2...").

**Audience:**

Graph generative models have recently become a very active research branch. Therefore, the findings of this paper will be of interest to some individuals in TMLR's audience.

**Claims And Evidence:**

The paper claims that existing score-based models for graph generation either completely disregard edge features or use some diffusion process for them and a different one for node features and other graph elements. The paper presents a joint score-based model which can generate simultaneously both node and edge attributes. The proposed architecture seems reasonable while the main claims are backed up by the reported empirical results.